# Biogenic cloud nuclei in the Central Amazon during the transition from wet to dry season

James D. Whitehead[1], Eoghan Darbyshire[1], Joel Brito[2], Henrique M. J. Barbosa[2], Ian Crawford[1], Rafael Stern[3], Martin W. Gallagher[1], Paul H. Kaye[4], James D. Allan[1], Hugh Coe[1], Paulo Artaxo[2], and Gordon McFiggans[1]

[1]Centre for Atmospheric Science, SEAES, University of Manchester, Oxford Road, Manchester, M13 9PL, UK
[2]Institute of Physics, University of São Paulo, Rua do Matão, Travessa R, 187. CEP 05508-090, São Paulo, S.P., Brazil
[3]Instituto Nacional de Pesquisas da Amazônia, Av. André Araújo, 2936, Aleixo, CEP 69060-001, Manaus, AM, Brazil
[4]Science and Technology Research Institute, University of Hertfordshire, AL10 9AB, UK

*Correspondence to:* G. McFiggans
(g.mcfiggans@manchester.ac.uk)

**Abstract.** The Amazon basin is a vast continental area in which atmospheric composition is relatively unaffected by anthropogenic aerosol particles. Understanding the properties of the natural biogenic aerosol particles over the Amazon rainforest is key to understanding their influence on regional and global climate. While there have been a number of studies during the wet season, and of biomass burning particles in the dry season, there has been relatively little work on the transition period - the start of the dry season in the absence of biomass burning. As part of the Brazil-UK Network for Investigation of Amazonian Atmospheric Composition and Impacts on Climate (BUNIAACIC) project, aerosol measurements, focussing on unpolluted biogenic air masses, were conducted at a remote rainforest site in the Central Amazon, during the transition from wet to dry seasons, in July, 2013. This period marks the start of the dry season, but before significant biomass burning occurs in the region.

Median particle number concentrations were 266 cm$^{-3}$, with size distributions dominated by an accumulation mode of 130 - 150 nm. During periods of low particle counts, a smaller Aitken mode could also be seen around 80 nm. While the concentrations were similar in magnitude to those seen during the wet season, the size distributions suggest an enhancement in the accumulation mode compared to the wet season, but not yet to the extent seen later in the dry season, when significant biomass burning takes place. Submicron non-refractory aerosol composition, as measured by an Aerosol Chemical Speciation Monitor (ACSM), was dominated by organic material (around 81%). Aerosol hygroscopicity was probed using measurements from a Hygroscopicity Tandem Differential Mobility Analyser (HTDMA), and a quasi-monodisperse Cloud Condensation Nuclei counter (CCNc). The hygroscopicity parameter, $\kappa$, was found to be low, ranging from 0.12 for Aitken mode particles to 0.18 for accumulation mode particles. This was consistent with previous studies in the region, but lower than similar measurements conducted in Borneo, where $\kappa$ ranged 0.17 - 0.37.

A Wide Issue Bioaerosol Sensor (WIBS-3M) was deployed at ground level to probe the coarse mode, detecting primary biological aerosol by fluorescence (Fluorescent Biological Aerosol Particles, or FBAP). The mean FBAP number concentration was 400 ± 242 l$^{-1}$, however this ranged from around 200 l$^{-1}$ during the day to as much as 1200 l$^{-1}$ at night. FBAP dominated the coarse mode particles, comprising between 55% and 75% of particles during the day to more than 90% at night. Non-FBAP

did not show a strong diurnal pattern. Comparison with previous FBAP measurements above canopy at the same location suggests there is a strong vertical gradient in FBAP concentrations through the canopy. Cluster analysis of the data suggests that FBAP were dominated (around 70%) by fungal spores. Further, long-term measurements will be required in order to fully examine the seasonal variability, and distribution through the canopy of primary biological aerosol particles.

5    This is the first time that such a suite of measurements has been deployed at this site to investigate the chemical composition and properties of the biogenic contributions to Amazonian aerosol during the transition period from the wet to dry seasons, and thus provides a unique comparison to the aerosol properties observed during the wet season in previous, similar campaigns. This was also the first deployment of a WIBS in the Amazon rainforest to study coarse mode particles, particularly primary biological aerosol particles, which is likely to play an important role as ice nuclei in the region.

## 1    Introduction

The Amazon Basin consists of the world's largest rainforest, covering an area of 5.5 million square kilometres. The Amazon rainforest is one of the few continental regions where atmospheric processes are minimally influenced by anthropogenic emissions, particularly during the wet season, and ambient conditions can represent, to some extent, those of the pristine pre-industrial era (Pöschl et al., 2010). Concentrations and properties of aerosol particles are largely governed by biogenic emissions, of both primary biological aerosol particles (PBAP) and biogenic volatile organic compounds (BVOC), which contribute to secondary organic aerosol (SOA). On a regional scale, in the wet season, the hydrological cycle is strongly influenced by these biogenic aerosol emissions, which provide most of the cloud condensation nuclei and thereby influence the radiation balance and cloud lifetime (Pöschl et al., 2010). In the dry season, by contrast, widespread biomass burning can result in a substantially increased aerosol optical depth over large areas of Amazonia, as well as modified cloud properties and suppressed precipitation (Andreae et al., 2004).

Previous studies in the pristine Amazon rainforest showed that fine particles (which account for most of the cloud condensation nuclei), consist mostly of secondary organic material derived from oxidised biogenic gases (Pöschl et al., 2010; Martin et al., 2010a; Allan et al., 2014; Chen et al., 2015). A lack of evidence for new particle formation during ground-based measurements (Zhou et al., 2002; Rissler et al., 2004; Martin et al., 2010a) implies that nucleation processes occur at higher altitudes, and new particles are entrained into the boundary layer from aloft (Krejci et al., 2005; Martin et al., 2010b). Larger super-micron particles are dominated by primary biological aerosol particles (PBAP) released from rainforest biota (Elbert et al., 2007; Pöschl et al., 2010; Huffman et al., 2012), which can play a significant role as ice nuclei (Prenni et al., 2009). These PBAP consist of wind-driven particles, such as pollen, bacteria, and plant debris, as well as actively ejected material, such as fungal and plant spores. Non-biological particles observed in the Amazon in the super-micron size range largely consist of advected Saharan dust and sea-salt from the Atlantic (Formenti et al., 2001; Worobiec et al., 2007; Martin et al., 2010b).

The low aerosol number concentrations in the pristine Amazon rainforest (typically a few hundred $cm^{-3}$) mean that CCN activation in convective clouds is often aerosol limited (Pöschl et al., 2010). It is clear that there is a strong coupling between

the rainforest biosphere and the hydrological cycle in the Amazon Basin, with biogenic aerosol particles providing the nuclei for clouds, which in turn sustain the rainforest through precipitation (Pöschl et al., 2010).

Improving our knowledge of these processes is necessary to understanding the influence the Amazon rainforest has on regional and global climate and atmospheric composition, and how changing land use and climate in Amazonia will impact on this (Artaxo et al., 2013). To this end, the Brazil-UK Network for Investigation of Amazonian Atmospheric Composition and Impacts on Climate (BUNIAACIC) was established to define and nurture a framework within which future UK contributions to studies in these areas may be coordinated. As part of the BUNIAACIC project, a short-term intensive measurement campaign was undertaken at a pristine rainforest site in July, 2013. The main focus of this study was to look at natural (biogenic) aerosol at this site at the beginning of the dry season (also referred to as the transition fron wet to dry season), and to compare to previous measurements made during the wet season at the same location (Martin et al., 2010a). Here we present the results of this study.

## 2 Methodology

### 2.1 Measurement Site and Sampling

The measurements were conducted at a remote site in pristine Amazonian rainforest between the 4th and 28th July 2013, during the transition from the wet to dry seasons. This is around the start of the dry season, but before significant biomass burning takes place. In July 2013, the total rainfall measured was 153 mm, mostly concentrated at the start and end of the month (during the measurement period itself, the rainfall was 77 mm). For the purposes of comparison, the AMAZE-08 campaign, which was conducted at the same site, had 370 mm of rainfall over the course of 5 weeks during the wet season (Martin et al., 2010a). In this study, the quartile ranges in temperature were 24°C - 29°C during the daytime, and 23°C - 25°C at night; relative humidity ($RH$) was 72% - 92% by day and 85% - 96% at night. By contrast, the conditions during AMAZE-08 were cooler and more humid, with temperature ranging 23°C - 27°C during the day and 22°C - 24°C at night; $RH$ ranging 88% - 99% by day and 96% - 100% at night (Martin et al., 2010a).

Sampling was done at the TT34 tower (2°35'40"S 60°12'33"W, elevation 110 m), in the Reserva Biológica do Cuieiras, approximately 60 km NNW of the city of Manaus in Brazil (see Fig. 1). The site is representative of near-pristine conditions, and no biomass burning takes place within the reservation, however the site can be affected by regional transport of pollutants including emissions from Manaus and biomass burning (Artaxo et al., 2013; Rizzo et al., 2013). Locally, accommodation for researchers and a 60 kW diesel generator were situated 0.33 km and 0.72 km, respectively, in a WNW direction from the tower. Intensive measurement campaigns have taken place at this site in the past (e.g. Martin et al., 2010a), and long term measurements have been conducted since 2008 (Artaxo et al., 2013; Rizzo et al., 2013). During this experiment, local time was 4 hours behind UTC.

A laminar sample flow of about 17 lpm was drawn through a 3/4" OD stainless steel line from a height of 39 m (about 10 m above canopy height) down to a ground level air conditioned container, in which the instruments were housed. Before entering the container, the sample was passed through an automatic regenerating adsorption aerosol dryer (Tuch et al., 2009). This kept

the $RH$ in the sample flow to between 20% and 40%. For the range of flows rates during this campaign the transmission range has previously been calculated from 4 nm to 7 μm (Martin et al., 2010a). Instruments drawing off this dried sample flow included a Hygroscopicity Tandem Differential Mobility Analyser (HTDMA; University of Manchester), and a Cloud Condensation Nuclei counter (CCNc; CCN-100, Droplet Measurement Technologies). Upstream of these instruments, the sample flow (2 lpm) was further dried to an $RH$ of between 15% and 25% with a nafion dryer operating with a counterflow of dry compressed air. The flow then passed through an electrical ionizer (model 1090, MSP Corporation), providing a charge-neutralised aerosol sample to the instruments. These same instruments were deployed in Borneo during the OP3 project (Irwin et al., 2011). Further details of the HTDMA and CCNc are given below.

Core instruments running at the site, on the same inlet, included a Multi Angle Absorption Photometer (MAAP; model 5012, Thermo-Scientific), a Condensation Particle Counter (CPC; model 3772, TSI), and an Aerosol Chemical Speciation Monitor (ACSM; Aerodyne Research Inc.). The ACSM was used to measure mass concentrations of particulate ammonium, nitrate, sulphate, chloride, and organic species in the submicron size range. Mass calibration was obtained by sampling mono disperse ammonium nitrate and ammonium sulphate. The instrument collection efficiency was calculated to be 1 during BUNIAACIC, through the comparison of the mass concentration of species measured by the ACSM and MAAP (black carbon equivalent; $BC_e$) with the integrated mass of the SMPS. Further instrumental details and data post-processing is given by Brito et al. (2014) and Stern et al. (in preparation). A weather station (Davis, USA) at the top of the tower provided meteorological data (wind speed and direction, temperature, $RH$, etc.).

As well as the instruments in the container, a Wide Issue Bioaerosol Sensor (WIBS; model 3M, University of Hertfordshire) was operated in a weatherproof box on the ground, a short distance from the base of the tower, with a short (1 m) 1/4" OD stainless steel inlet (more details are provided below). Other core instruments running at the site, but not used in this study, are detailed by Artaxo et al. (2013).

## 2.2 HTDMA measurements

In the HTDMA (Cubison et al., 2005; Good et al., 2010), a dry aerosol sample is mobility size-selected with the first DMA and then humidified to a set $RH$. The second DMA is then used to measure the size distribution of the humidified aerosol, to give the distribution of Growth Factor (defined as the ratio of humidified to dry aerosol diameter: $D/D_0$) as a function of $RH$ and dry diameter ($GF_{RH,D_0}$). Quality assurance and inversion of the data was performed using the TDMAinv toolkit of Gysel et al. (2009). During normal operation, the first DMA cycled through 5 mobility sizes (45 nm, 69 nm, 102 nm, 154 nm and 269 nm; calibrated values), and the monodisperse flow after the first DMA was humidified to a target $RH$ of 90%. The $RH$ measured in DMA2 remained fairly stable ($\pm 2\%$) for most of the measurement period, and the variation was accounted for by correcting the data to the target $RH$ within the inversion toolkit (Gysel et al., 2009). In addition to this normal mode of operation, humidograms were run on the 21st and 23rd July. In this mode, cycling through 3 dry sizes (45 nm, 102 nm and 269 nm), the $RH$ in the second DMA was gradually varied between 45% and 95% in order to determine how the $GF$ of ambient aerosol varies with $RH$.

In both DMAs, a ratio of 10:1 was maintained between the sheath and sample flows, and these were calibrated using an air-flow calibrator (Gillibrator-2, Sensidyne). The first DMA was size calibrated at the start of measurements using NIST-traceable polystyrene latex spheres (PSL; Fisher Scientific), sizes 100, 150, 200 and 300 nm, nebulised with an aerosol generator (model ATM 226; TOPAS). Dry scans (in which the sample is not humidified between the DMAs) were run on an approximately

weekly basis in order to monitor the size offset between the two DMAs and to define the width of the DMA transfer functions (Gysel et al., 2009). The HTDMA was further verified by sampling nebulised ammonium sulphate, monitoring the growth factors for a range of $RH$ (68% to 92%) at a given size (140 nm), and comparing to modelled values (ADDEM; Topping et al., 2005). More details of the calibration procedures for this instrument are given by Good et al. (2010).

### 2.3 CCNc measurements

The CCNc (Roberts and Nenes, 2005) operated downstream of a DMA (model 3081, TSI), the voltage of which was controlled with a classifier (TSI, model 3080) stepping discretely through a mobility size range 16 nm to 325 nm. This quasi-monodisperse aerosol sample flow was then split isokinetically between the CCNc and a CPC (TSI, model 3010). The flow into the CPC was further diluted with filtered air by a factor of 2 in order to match the flow into the CCNc. Inside the CCNc, the aerosol flowed through a wetted column with a temperature gradient, providing supersaturated conditions in which a proportion of the particles

activated and were detected by an Optical Particle Counter (OPC) at the bottom of the column. Throughout the deployment, the CCNc cycled through 5 calibrated supersaturation setpoints: 0.15%, 0.26%, 0.47%, 0.80% and 1.13%. The ratio of activated particles to total particles (measured by the CPC), can be determined as a function of dry particle diameter and supersaturation (the activated fraction: AF). By fitting a sigmoid curve function to this activation spectrum, the dry diameter at which 50% of particles activate ($D_{50}$) was derived. The hygroscopicity parameter, $\kappa$ (Petters and Kreidenweis, 2007), was then derived

from $D_{50}$ and supersaturation using the $\kappa$-Köhler model. In addition, the total number of CCN ($N_{CCN}$) was calculated by integrating the number size distribution above $D_{50}$.

The DMA was calibrated using PSLs of the same sizes as with the HTDMA. The CCNc was calibrated by flowing nebulised ammonium sulphate into the system and determining the supersaturation at which 50% of the particles of a given dry size activate. This critical supersaturation is then compared to modelled values (ADDEM; Topping et al., 2005) to determine the

slope and offset.

### 2.4 Bio-aerosol measurements

Fluorescent Biological Aerosol Particles (FBAP) in the optical size range $0.5 \leq D_p \leq 20$ μm were detected using the WIBS-3M (Kaye et al., 2005; Foot et al., 2008; Stanley et al., 2011), which operates on the principle of ultraviolet light induced fluorescence of molecules common to most biological material, specifically Tryptophan and the co-enzyme NADH. Two se-

quential pulses of UV light are provided by filtered Xenon lamps at 280 nm and 370 nm to excite Tryptophan and NADH, respectively. Fluorescence is then detected in the ranges 310–400 nm and 400–600 nm following the Tryptophan excitation, and 400–600 nm following the NADH excitation (i.e. 3 fluorescence channels; FL1, FL2, and FL3, respectively). In addition,

the WIBS-3M provides a dimensionless particle assymmetry factor ($A_f$) as a proxy for particle morphology, as detailed by Crawford et al. (2015). Particles smaller than 0.8μm were rejected from analysis due to low counting efficiency.

The baseline fluorescence of the instrument is measured during so-called forced trigger (FT) sampling periods, where the instrument triggers the flash lamps and records the resultant fluorescence in the absence of aerosol in the sample volume. The mean fluorescence in a FT period is treated as the baseline fluorescence of the optical chamber during the sample period. For a particle to be considered fluorescent (FBAP) it must exhibit a fluorescence greater than a threshold value, defined as the baseline fluorescence plus 3 standard deviations, in any channel. During data processing the threshold value for each channel is subtracted from the single particle fluorescence data and the value is clipped at zero with all values greater than zero being considered significantly fluorescent compared to the instrument baseline. All reported fluorescence measurements are relative to the applied threshold and not the absolute detector intensities. This is consistent with previous studies using this instrument (Robinson et al., 2013; Crawford et al., 2014, 2015, 2016), and a detailed description of this data processing method is provided by Crawford et al. (2015). The thresholds remained consistent over 58 FT periods throughout the measurements at: 112.4 ± 3.9 for channel FL1, 284.6 ± 7.8 for FL2, and 164.6 ± 5.7 for FL3. The ambient threshold determination method (Perring et al., 2015) was not used here due to the majority of particles being fluorescent in nature.

Size calibration of the WIBS-3M consisted of using PSLs with a physical diameter of 1.0 μm. Blue fluorescent latex spheres (1.0 μm diameter; Thermo Scientific) were also used to ensure that the excitation and fluorescence channels were operating correctly. The WIBS-3M inlet was operated at a total flow rate of 2.3 lpm (±5%). 90% of this was directed through a HEPA filter and used as a sheath flow, constraining the remaining 0.23 lpm for the scattering chamber sample flow from which particle concentrations were derived.

Particles with fluorescent magnitudes above the threshold are termed FBAP, as they represent a lower limit of PBAP, some of which may not be detected by this method if their fluorescence goes undetected, or they simply don't fluoresce (Gabey et al., 2010; Huffman et al., 2012). Non-biological fluorescent material can also be detected by the WIBS should its excitation and emission profile match that of the instrument. Generally the identified interferents are smaller than the detection limit of the instrument. Polycyclic aromatic hydrocarbons (PAH) such as naphthalene, and soot containing PAHs have been shown to fluoresce in FL1 (Pöhlker et al., 2012; Toprak and Schnaiter, 2013), however they would not be expected to be seen in significant concentrations outside of the pollution events at such a remote site. Mineral dusts contain a small subset of fluorescent aerosol within their population (≈10%), and given their ubiquitous nature may present a significant source of interferents to the UV-LIF method (Toprak and Schnaiter, 2013). Their observed fluorescent intensity, however, is considerably weaker than is observed for biofluorophores (Pöhlker et al., 2012), and if they were present in any significant concentrations they would likely form their own cluster in the cluster analysis discussed in section 3.4 (Crawford et al., 2016). It should also be noted that the technique does not distinguish between biological particles and fluorescent material attached to non-biological particles (e.g. dust).

This instrument has previously been deployed in Borneo, and further details of its operation are given by Gabey et al. (2010). In this experiment, the instrument was positioned in a small clearing, a few metres away from the rainforest understorey. It should be noted that the WIBS-3M measurements only ran until the 10th July, and so did not overlap with the other principle measurements (HTDMA, CCNc, ACSM), which began on the 10th July. Meteorological conditions were fairly consistent over

the whole measurement period, and so all measurements discussed here are considered representative of the same general period (i.e. the transition from wet to dry season).

## 2.5 Removal of pollution episodes

In order to focus on the natural (biogenic) aerosol, for comparison with the wet season, it was necessary to exclude periods affected by pollution. While the site is described as pristine, it can nevertheless be affected by local emissions and regional transport of pollutants: biomass burning emissions from outside the reserve; the urban plume from Manaus; and pollution from the nearby diesel generator.

For each day of the campaign 7-day back trajectories were calculated using the HYSPLIT model (Draxler and Hess, 1998) at 30 minute intervals and 6 altitudes above TT34 (0, 250, 500, 1000, 2000 and 4000 m.a.s.l). The horizontal and vertical wind fields employed here were from the NCEP/NOAA 1°x 1°Global Data Assimilation System (GDAS) reanalysis product. These back trajectories were used to identify air masses arriving at TT34 which had either passed over Manaus or passed nearby active fire zones. The results were largely the same between 0 and 2000 m, with very little influence from the upper level flow at 4000 m.

A bounding box was drawn between -3.16°to -2.88°latitude and -60.12°to -59.81°longitude to define the Manaus influence zone (see Fig. 1), and any back trajectory passing over this box at any altitude up to 2000 m was flagged. Air masses potentially impacted by biomass burning were identified by coupling the back trajectory measurements to satellite detected fires as measured by the MODIS instrument. This operates on the Aqua and Terra satellites, which have local overpass times in the morning and afternoon respectively. The fire detection data (specific product: MCD14ML) was produced by the University of Maryland and acquired from the online Fire Information for Resource Management System (FIRMS; https://earthdata.nasa.gov/data/near-real-time-data/firms/about). At each location along the back trajectories the surrounding 1°box was interrogated for any fire counts at the nearest terra/aqua overpass. If any were present this trajectory was flagged as potentially influenced by biomass burning. This technique is subject to uncertainties associated with trajectory errors (e.g. Fleming et al., 2012) and the detection of fires in cloudy scenes, or false detection of fires (Schroeder et al., 2008), and therefore can only be considered qualitative. Finally, data were investigated for possible contamination from the generator if the local wind direction was in the range 270°-340°, however there were no instances of generator contamination during the measurement periods of this study.

In the event of any flag, the black carbon data (from the MAAP) were checked along with the particle counts (where available), and aerosol data were excluded if the pollution flag coincided with a significant increase in these concentrations. No other increases in black carbon concentrations were seen outside the flagged periods. Approximately 28% of the HTDMA and CCNc data were removed in this way, with 5% of the data being flagged as possibly impacted by biomass burning and most of the rest due to the Manaus urban plume. The ACSM, which was not necessarily operating at the same times as the HTDMA and CCNc, had approximately 9% of its data removed due to pollution flags (almost entirely due to the urban plume from Manaus).

## 3 Results and Discussion

### 3.1 Size distributions

The particle number size distribution recorded over the measurement period of this study can be seen in Fig. 2. This shows a broad accumulation mode peak at 130 - 150 nm with a median number concentration of 266 cm$^{-3}$ (calculated from the integral of the size distribution curve). Despite observing aerosol number concentrations comparable to previous observations during the wet season, the shape of the distribution resembles those measured in the dry season, although the concentrations during the latter are considerably higher at 2200 cm$^{-3}$ (Artaxo et al., 2013). Figure 3 shows the size distribution again combined with the coarse mode FBAP measurements from the WIBS. In terms of number concentration, the submicron modes dominate the coarse mode by a factor of $10^3$. The WIBS measurements are discussed in further detail below.

The size distribution, however, was quite variable over the period of the measurements, as can be seen in the time-series in Fig. 4, and varied with total particle number concentrations. Median size distributions observed when the total number concentration was above or below 200 cm$^{-3}$ are shown in Fig. 5. During periods of low particle counts, an Aitken mode is also seen, with a mode around 80 nm, while the size distribution during episodes of higher concentrations is dominated by the accumulation mode, possibly masking the smaller mode. Such a size distribution profile, dominated by accumulation mode aerosols, has also been reported during the dry season in western Amazonia, in the deforestation arc, during biomass burning events (Brito et al., 2014), albeit with substantially higher concentrations.

### 3.2 Composition

Submicron non-refractory aerosol composition, as measured by the ACSM during the period of this study, is illustrated in Fig. 6. The mean mass loadings for organic material, sulphate and nitrate were $2.13 \pm 0.75$ µg m$^{-3}$, $0.11 \pm 0.04$ µg m$^{-3}$, $0.08 \pm 0.03$ µg m$^{-3}$, respectively ($\pm$ 1 standard deviation). Organic material dominated the submicron aerosol, comprising around 81% of the total mass (86% of non-refractory material), on average. Such a high fraction of organics compares well with previous observations in the Amazon basin (Artaxo et al., 2013; Brito et al., 2014; Andreae et al., 2015). BC$_e$ concentrations are also shown, with a mean mass loading of $0.25 \pm 0.01$ µg m$^{-3}$. This is consistent with previous wet season measurements in the Amazon (Artaxo et al., 2013; Andreae et al., 2015).

The mass fractions of non-refractory aerosol and BC$_e$ are shown in the bottom panel of Fig. 6. Due to the noise in the ammonium signal (see Fig. 6), resulting from concentrations below the limit of detection of 0.3 µg m$^{-3}$ for the ACSM, it was necessary to estimate the ammonium from the nitrate and sulphate mass loadings for the purpose of mass fraction calculations. The time-series of mass fractions show that, while the mass loadings vary considerably, particularly organics, the composition is relatively consistent as a proportion of the aerosol mass. Organic mass fractions remain steady around 81% of the total mass, until the 22nd and 23rd July, when a slight increase in BC$_e$ is seen.

Levoglucosan, a major constituent of biomass burning aerosol, fragments in AMS and ACSM instruments at a mass-to-charge ratio ($m/z$) of 60 (Alfarra et al., 2007), and so the fraction $f_{60}$ is frequently used as a marker for biomass burning (Artaxo et al., 2013; Chen et al., 2009). The mean $f_{60}$ from the ACSM data in this study, after removal of pollution episodes,

was 0.19% ± 0.07%. This is well below 0.3%, which is considered to be the upper limit for background air masses not affected by biomass burning (Cubison et al., 2011). It should be noted that previous studies in the Amazon have observed that a large fraction of the biomass-burning related organic aerosols do not present a significant $f_{60}$ signal, due to long-range transport (Brito et al., 2014). It can be said, however that the relatively low $f_{60}$ observed here suggests that, on average, these

measurements were not strongly impacted by local biomass burning emissions.

Previous studies have successfully identified FPAB markers on ambient aerosol in the Amazon using an aerosol mass spectrometer (Schneider et al., 2011), a method which relies strongly on the high-resolution capabilities of the instrument used at the time. Given the unity mass resolution of the ACSM, similar methodology has not been applied here.

### 3.3    Aerosol water uptake

The HTDMA ran from the 13th to the 28th July. Figure 7 shows the time-series of $RH$-corrected $GF$ distributions for all dry sizes, as derived from the HTDMA data using the TDMAinv toolkit. These largely exhibit a single mode at each size, roughly in the range of 1.2 to 1.4. Some variability can be seen, for example on the 21st and 23rd July, but for the most part, peak growth factors remained relatively stable over the measurement period. This is consistent with the stable mass fractions seen in the composition data from the ACSM (see Fig. 6, bottom panel). The variability and slight decrease in $GF$ at some sizes

on the 22nd and 23rd July may also be attributed to the slight increase in the mass fraction of $BC_e$ (Fig. 6). High peak growth factors ($> 1.6$) can briefly be seen on the night of the 15th July, shortly before a pollution event, however without composition data available on that day (Fig. 6), it is difficult to speculate as to the nature of this.

Smaller, more hygroscopic ($GF > 1.5$) modes can be seen at the lower dry diameters, while the larger particles also show a hydrophobic mode in the growth factor distribution. The contribution of the hydrophobic mode to the larger particles is

small ($< 10\%$ in number) and may be due to some unknown local anthropogenic influence that was not accounted for. The averages of the growth factor at the peak of the growth factor distribution (i.e. the dominant mode) are shown in table 1 and Fig. 2. They show an increase with dry diameter, reflecting the difference between Aitken and accumulation mode aerosol: organic mass fractions are highest in the Aitken mode, while elevated sulphate mass fractions have been previously seen in the accumulation mode (Gunthe et al., 2009; Pöschl et al., 2010). It should be noted, however, that the elevated sulphate events

observed by Gunthe et al. (2009) were likely linked to long-range transport of biomass burning aerosol from Africa, which, due to a combination the African burning season and large scale circulation, tends to impact the Amazon forest more often during the wet season (Ben-Ami et al., 2010).

The campaign averages of the CCNc derived parameters, $D_{50}$, $\kappa$ and $N_{CCN}$ are given for each set supersaturation in table 2. The $\kappa$ values are also plotted against $D_{50}$ in Fig. 2. Consistent with the growth factor data, and with previous measurements

at this site (Gunthe et al., 2009), they show more hygroscopic particles at larger diameters ($\kappa \approx 0.12$ below 100 nm, and $\kappa \approx 0.18$ around the accumulation mode).

Reconciliation between sub- and super-saturated particle water uptake for these measurements has already been investigated by Whitehead et al. (2014). They showed that there was agreement within the variability of the data, with a slightly underestimated hygroscopicity from the HTDMA data compared to the CCNc at lower supersaturations (larger dry diameters). The

analysis of Whitehead et al. (2014) considered the full dataset without separating out the pollution events, however performing the same analysis on the 'clean' data did not result in any significant difference.

## 3.4 FBAP measurements

Measurements of biological particles in the Amazon are important as they are considered to have a strong influence on clouds as ice nuclei (Pöschl et al., 2010). The WIBS-3M operated uninterrupted from the morning of the 3rd July until 10th July. The mean particle number concentration measured by the WIBS-3M during this period was $464 \pm 250 \, \mathrm{l}^{-1}$ (1 standard deviation), while the mean FBAP number concentration was $400 \pm 242 \, \mathrm{l}^{-1}$ (i.e. accounting for 86% of the particles in the size range of the instrument). The time-series of number concentrations for the duration of this period is shown in Fig. 8. This shows coarse mode particles were dominated by FBAP number concentrations, which exhibited a strong diurnal cycle with concentrations varying from around $200 \, \mathrm{l}^{-1}$ during the daytime up to as much as $1200 \, \mathrm{l}^{-1}$ at night. The diurnal variation (Fig. 9) shows that FBAP number concentrations plateaued from around 21:00 through the night, began to drop from 05:00, reached a minimum by 11:00 and started increasing again from 15:00. The FBAP fraction was highest (more than 90%) at night, and remained high until around 08:00 - even after FBAP number concentrations began decreasing. This dropped to between 55% and 75% during the day, helped in part by an apparent increase in non-FBAP concentrations, before steadily increasing in line with the FBAP concentrations through the late afternoon / early evening.

There are a number of factors driving the diurnal cycle in coarse mode particles, as discussed by Huffman et al. (2012). Previous studies at this and a nearby site, utilizing electron and light microscopy, have identified the FBAP as predominantly fungal spores (Graham, 2003; Huffman et al., 2012). Similar diurnal cycles have been seen in airborne fungal spore densities at other tropical rainforest locations (Gilbert and Reynolds, 2005; Elbert et al., 2007). The observed night-time peak in FBAP number concentrations in Fig. 9 is consistent with nocturnal sporulation driven by increasing $RH$ (see bottom panel; note that $RH$ is measured above the canopy). The dependence of fungal spore release on meteorological conditions, however, varies greatly according to species, and any relationship is non-trivial (Jones and Harrison, 2004). FBAP number concentrations begin dropping several hours before any decrease in $RH$, and the FBAP fraction also remains high (Fig. 9). This suggests that the morning decrease in FBAP is not necessarily due to a cessation of emission processes, but may also be the result of a break-up of the nocturnal boundary layer around sunrise (Whitehead et al., 2010; Huffman et al., 2012). Graham (2003) and Huffman et al. (2012) suggest that the night-time increase in coarse mode particles is due, at least in part, to the shallow nocturnal boundary layer. The slight increase in non-FBAP concentrations during the day may be a result of enhanced particle exchange through the canopy, facilitated by sporadic turbulent events, as described by Whitehead et al. (2010), bringing non-FBAP that had originated elsewhere into the space below canopy.

Figure 10 shows the number size distributions reported by the WIBS-3M during the measurement period. Again, FBAP clearly dominates the particle number concentrations for $D_p > 1 \, \mu\mathrm{m}$, however non-FBAP concentrations are higher for particles smaller than $1 \, \mu\mathrm{m}$ measured by the WIBS-3M (i.e. down to the instruments 50% detection diameter of $0.8 \, \mu\mathrm{m}$). The FBAP number size distribution shows a peak at around $1.8 \, \mu\mathrm{m}$, while the non-FBAP distribution is characterized by a flatter, broader peak between 0.8 and $1.3 \, \mu\mathrm{m}$. Non-fluorescent particles at this site have previously been identified as mineral dust,

non-fluorescent biological aerosol, and inorganic salts (Huffman et al., 2012). Caution must be applied when interpreting the sub-micron fluorescent aerosol fraction due to the reduced fluorescent counting efficiency for particles $D_p < 0.8$ μm (Gabey et al., 2011), which may lead to an underestimation of the fluorescent aerosol fraction at small sizes.

A Ward linkage cluster analysis using the z-score normalisation was applied to the data, where the optimum number of retained distinct clusters was determined using the Calinski–Harabasz criterion. Prior to analysis, all non-fluorescent and saturated particles, and particles smaller than 0.8 μm in diameter were excluded, resulting in approximately 15% of the single particle data being rejected. The asymmetry factor and size inputs were converted to log space prior to normalisation and clustering. Complete information on this technique is given by Crawford et al. (2015), who used the same instrument model in the same way as in this study. This analysis revealed three distinct fluorescent classes of particles (Cl1-3). The statistical parameters of each cluster are shown in table 3. It can be seen that Cl1 and Cl2 display similar characteristics; specifically, they mainly fluoresce in FL1 with weak fluorescence in the other channels, although the intensities are greater for Cl1, suggesting they are distict sub-classes. The two clusters correlate strongly ($r^2 = 0.86$) with each other, hence both have been combined in Fig. 9. They show similar fluorescent signatures to the clusters attributed to fungal spores by Crawford et al. (2014, 2015) based on comparison with other sampling techniques and the diurnal emission pattern. In this study, they show higher concentration overnight (Fig. 9), and a strong correlation to $RH$ (Fig. 11. Together, clusters Cl1 and Cl2 contribute approximately 70% to the total fluorescent particle concentration, regardless of time of day, suggesting that FBAP were dominated by fungal spores during this study. A third cluster, Cl3, shows very low concentrations (around 20 $l^{-1}$), with no strong diurnal trend, however there is insufficient data to speculate upon the nature of this cluster (such as response to rainfall). The asymmetry factor for each cluster was around 30, suggesting that the particles are aspherical in nature. A similar value of asymmetry factor was observed by Crawford et al. (2015) in the ambient fungal cluster, further suggesting that clusters Cl1 and Cl2 are representative of fungal spores.

PBAP classification via the comparison of single particle fluorescent signatures to laboratory samples is an ongoing area of research (e.g. Hernandez et al., 2016). Such direct comparison for this purpose is not possible here due to differences in the instruments used (i.e., different excitation/detection wavebands and optical chamber design). Even comparing results between the same model of instrument with identical detector/filter configurations has been difficult (Hernandez et al., 2016) due to the current lack of a robust fluorescence calibration method.

## 3.5 Comparison with previous studies

### 3.5.1 Submicron aerosol

Aerosol water-uptake studies have previously been conducted at the TT34 site by Gunthe et al. (2009) using size-selected CCNc measurements, and at Balbina (110 km NE of TT34) by Zhou et al. (2002) using a HTDMA, both during the wet season. HTDMA and CCNc measurements were also made at Balbina during the transition from wet to dry season by Rissler et al. (2004). In addition, HTDMA measurements from pasture-land in SW Amazonia at the end of the dry season / beginning of wet season are presented by Rissler et al. (2006) and Vestin et al. (2007). This study represents the first measurements with

HTDMA and monodisperse CCN instruments at TT34 during the transition from wet to dry seasons. Concurrent CCNc and HTDMA measurements have also been conducted in Borneo, SE Asia, by Irwin et al. (2011), providing a useful comparison with a different tropical rainforest region.

The HTDMA growth factor measurements of Zhou et al. (2002) showed a similar pattern to this study: a dominant mode of "less hygroscopic" particles ($GF \approx 1.16$ - $1.32$), accompanied at times by a hydrophobic mode ($GF < 1.06$; particularly at the larger particle sizes), and a more hygroscopic mode ($GF \approx 1.38$ - $1.54$). The growth factors of the less hygroscopic particles are compared in Fig. 12, along with the other studies (note that Rissler et al. (2004) define "less hygroscopic" as $GF \approx 1.17$ - $1.5$). All the measurements showed a similar increase in growth factor with dry diameter. The growth factor values from this study were slightly higher than those of Zhou et al. (2002) and Rissler et al. (2004), but the difference is within the variability of the measurements, and probably within the variability that has been seen between different HTDMA instruments (Duplissy et al., 2009; Massling et al., 2011). The "moderately hygroscopic" particles ($GF = 1.26$) observed by Rissler et al. (2006) exhibited growth factors in the same range as the other studies in Amazonia, however in this case, the hydrophobic mode ($GF \approx 1.05$ - $1.13$) was dominant for all but the larger particles ($> 135$ nm). Furthermore, strong diurnal cycles (daytime increases in the fraction of moderately hygroscopic particles) were observed (Rissler et al., 2006; Vestin et al., 2007), which were not seen during the current study. In contrast to the current study, the measurements of Rissler et al. (2006) and Vestin et al. (2007) were conducted in a region that has undergone heavy land use change and is strongly influenced by anthropogenic sources (Andreae et al., 2002), which may contribute to the observed diurnal pattern.

In contrast to the studies from Amazonia, aerosol growth factors measured in Borneo (Irwin et al., 2011) were somewhat higher: in the range 1.3 - 1.7 (Fig. 12). This can be explained by the fact that, while the site in Amazon benefited from a fetch of hundreds of kilometres of undisturbed rainforest, the site in Borneo was heavily influenced by marine air masses (Robinson et al., 2011). As discussed by Robinson et al. (2011), the sulphate loadings in Borneo were substantially higher than in Amazonia, even in air masses from across the island, which, with sulphate being more hygroscopic than organic aerosol, is a possible explanation for the higher growth factors.

The results of the humidogram are shown in Fig. 13, and compared to the humidogram data from Borneo (Irwin et al., 2011) and the humidogram fit for the wet season data of Rissler et al. (2006). Growth factors in Borneo were higher across the $RH$ range than in Amazonia. As with previous measurements, no deliquesence behaviour was seen in this study.

Values of $\kappa$ derived from the HTDMA and CCNc measurements during these studies are compared in Fig. 14, as a function of dry diameter. Here, the $\kappa$ from HTDMA measurements is derived using the average growth factor, rather than the peak of the less hygroscopic mode, for direct comparison with the CCNc derived values. The various measurements in Amazonia showed very similar $\kappa$, largely agreeing within the variability of the individual measurements. It can be said that water uptake measurements in Amazonia are consistent, and, as noted by Gunthe et al. (2009), show $\kappa$ to be around half that typically seen in other continental regions (Andreae and Rosenfeld, 2008).

The HTDMA derived $\kappa$ from the Borneo experiment shows more hygroscopic aerosol than in Amazonia, as discussed above, however the CCNc derived values are more in line with those in Amazonia. This discrepancy has been noted previously and possible reasons for it discussed by Irwin et al. (2011) and Whitehead et al. (2014). These were mainly related to differences

in the instrument setups and how they treat the aerosol. It should be noted that the discrepancy in the data from the Borneo experiment was the largest amongst a number of datasets studied by Whitehead et al. (2014), but the reason for this is not clearly understood.

In general, the particle concentrations and hygroscopic properties observed during this study were similar to those seen during wet season measurements in the Amazon rainforest. The main difference seen was that size distributions in this study were more strongly dominated by the accumulation mode: similar to those seen in the dry season (Artaxo et al., 2013), but in clean conditions with significantly lower number concentrations. Under these conditions, cloud droplet formation in convective clouds in this region is likely to be aerosol- limited (Reutter et al., 2009). Previous modelling studies have suggested this is the case during the wet season (Pöschl et al., 2010), in contrast to the dry season during periods of intense biomass burning when droplet number is largely controlled by the updraft velocity (Reutter et al., 2009).

In terms of composition, submicron non-refractory aerosol concentration during this experiment showed significantly higher concentration ($\approx$2.5 µg m$^{-3}$) than observed at the remote sites in Central Amazonia in previous years during the wet season, ranging from 0.4 µg m$^{-3}$ (Artaxo et al., 2013) and 0.6 µg m$^{-3}$ (Chen et al., 2009; Andreae et al., 2015). Conversely, the concentration is significantly lower than reported during the dry season (8.9 µg m$^{-3}$) (Andreae et al., 2015), due to this transitional period having no extensive biomass burning activities, albeit with already reduced wet deposition due to reduced precipitation. Interestingly, despite the marked changes in ambient concentration, very little difference is observed in terms of relative contributions considering this and previous studies, being strongly dominated by organics ($\approx$80%), followed by sulphate and minor contribution of nitrate and ammonium (Chen et al., 2009; Artaxo et al., 2013; Andreae et al., 2015).

It is not clear how meteorological conditions influence the differences between this study and those in the wet season. The warmer, dryer conditions of this study might result in more evaporation of SOA, whereas more precipitation in the wet season would lead to more washout, and therefore lower loadings. From the results, it would seem that the latter effect dominates, resulting in the higher organic loadings and accumulation mode aerosol seen in this study, but there may be other, more complex factors. Long-term measurements would be needed to fully investigate the influence of meteorology on particles in the Amazon.

### 3.5.2 Coarse mode aerosol

Huffman et al. (2012) conducted measurements of FBAP at the TT34 tower using an ultraviolet aerodynamic particle sizer (UV-APS) during the AMAZE-08 campaign. In contrast to this study, the AMAZE-08 measurements were taken during the wet season (February to March), from the top of the tower (i.e. above canopy). It is also worth comparing with the measurements of Gabey et al. (2010), who used the same WIBS-3 instrument to sample the aerosol above and below canopy in the rainforest of north-east Borneo.

The median number concentration of FPAB observed below the canopy in the current study was 363 l$^{-1}$, while the UV-APS measurements at the top of the tower by Huffman et al. (2012) were around a fifth of this, at 73 l$^{-1}$ (also median). In an intercomparison between the two different measurement techniques, Healy et al. (2014) found that, while there was agreement in total number concentrations, the counts in the fluorescence channels of the WIBS (particularly FL1) were substantially

higher than the UV-APS fluoresence counts, which would at least partly explain the difference here. The wetter, more humid conditions during the wet season measurement period of Huffman et al. (2012) would be expected to favour emission (Jones and Harrison, 2004; Zhang et al., 2015). On the other hand, the higher rainfall would also result in enhanced wet deposition during the wet season, especially above canopy. At other locations, Gabey et al. (2010) saw concentrations in Borneo often in excess of

$1500 \, l^{-1}$ below canopy, and around $200 \, l^{-1}$ above, using the same instrument at each site, while Gilbert and Reynolds (2005) observed substantially higher concentrations of fungal spores in the understorey than in the canopy during measurements in Queensland, Australia. Strong vertical gradients in biological particles are therefore regularly seen in rainforest environments, and would be an additional factor in the differences observed between the measurements at TT34. In a remote tropical rainforest in China, Zhang et al. (2015) estimated fungal spore concentrations to be around $50 \, l^{-1}$ based on chemical analysis of filters,

and found higher concentrations associated with rainfall events. A global modelling study by Spracklen and Heald (2014) found simulated surface annual mean concetrations of fungal spores to be around $100 \, l^{-1}$ over tropical forests (including Central Amazonia), which is consistent with this and other measurements at this site.

The fraction of FBAP in this study was, on average 85% of total coarse mode particles (and as much as 90%) whereas it was 24% in the AMAZE-08 campaign (41% in unpolluted conditions). The higher fraction at ground level would be expected, being

closer to the source, whereas above canopy, there is a stronger influence from non-fluorescent particles from external sources. Elbert et al. (2007) found fungal spores accounted for 35% of coarse mode particles, also in Central Amazonia, but their filter samples were taken at a pasture site adjacent to the rainforest. In Borneo, as in the Amazon, there was a higher fraction below the canopy (55%) than above (28%), however not as high as the 86% observed in this study. Reasons for this difference are unclear, but may include a stronger influence in Borneo of non-fluorescent particles from external sources, such as the nearby

coast. More consistent with the current study were the scanning electron microscopy (SEM) measurements reported by Pöschl et al. (2010) and Huffman et al. (2012), which attributed 80% of coarse mode particles to primary biological aerosol during the AMAZE-08 campaign, and also identified particles likely to be fungal spores.

One difference between the measurements of this study and others is the position of the mode in the FBAP number size distribution. Gabey et al. (2010) report the peak at 2.5 µm, while Huffman et al. (2012) observe the peak around 2.3 µm. By

contrast, the peak in this study was 1.8 µm. The difference between the two measurements at TT34 is likely due to the different measurement techniques, with the UV-APS found to be less sensitive to smaller fluorescent particles (Healy et al., 2014).

Diurnal variations between this study and that of Huffman et al. (2012) were similar, however Gabey et al. (2010) reported an additional increase in FBAP number concentrations in the afternoon in Borneo. This increase coincided with a peak in $RH$, and it is believed that this is linked (Gabey et al., 2010). In this study, the $RH$ increased more gradually through the afternoon

and evening (see Fig. 9, bottom panel), which may explain the lack of afternoon peak in FBAP compared to the Borneo results. Huffman et al. (2012) also do not observe a mid-afternoon peak in FBAP.

## 4 Conclusions

Measurements of aerosol concentrations and water uptake properties were conducted at a remote site in pristine Amazonian rainforest in July, 2013, during the transition from the wet to dry seasons. Back trajectories and wind sectors were examined in conjunction with black carbon concentrations in order to exclude any pollution episodes and ensure the aerosol measured were representative of background aerosol over the rainforest.

With any pollution episodes removed from the data, particle concentrations were low, with a median of 266 cm$^{-3}$. The particle size distributions were largely dominated by an accumulation mode around 130 - 150 nm, with a smaller Aitken mode apparent during periods of lower particle counts. Based on previous measurements contrasting wet and dry seasons (Artaxo et al., 2013), the results here may reflect the transition between the two seasons, with periods consistent with each at different times (but without considering any influence from biomass burning).

Aerosol chemical composition, as measured with an ACSM, was dominated by organic material, comprising around 81% of the total mass of non-refractory aerosol and BC$_e$. The mass fraction of organics was relatively consistent over the measurement period.

Aerosol water uptake and hygroscopicity was measured using an HTDMA and a CCNc. Good agreement was found between the measurements of both instruments. Particle growth factors from the HTDMA varied little over most of the measurement period and were typically between 1.2 and 1.4 (low hygroscopicity mode). Aerosol hygroscopicity was found to be low ($\kappa$ = 0.12) for Aitken mode particles, and increased slightly to $\kappa$ = 0.18 for accumulation mode particles. This is consistent with previous measurements at, or near this site, and with the observation that Aitken mode particle composition is dominated by organic material, while accumulation mode particles exhibited higher sulphate mass fractions (Pöschl et al., 2010).

Particles in the size range $0.5 \leq D_p \leq 20$ µm were measured using the WIBS-3M, which distinguishes fluorescent (representing a subset of primary biological aerosols, or FBAP) and non-fluorescent. FBAP dominated the coarse mode aerosol, accounting for as much as 90%. Concentrations of FBAP followed a strong diurnal cycle, with maximum concentrations during the night. This is likely driven by a combination of the dependence of emission processes on meteorological conditions and the diurnal cycle of the boundary layer.

The results from this study were also compared to measurements conducted in Borneo in 2008 (Irwin et al., 2011; Gabey et al., 2010; Robinson et al., 2011), contrasting the vast 'Green Ocean' of the Amazon rainforest to the island rainforest geography of SE Asia. In the submicron range, aerosol hygroscopicity was greater in Borneo, possibly due to the stronger marine influence of that region (Irwin et al., 2011). Coarse mode particles at both locations were dominated by FBAP (probably mostly fungal spores). Below canopy, the Amazon exhibited a higher fraction of FBAP than Borneo, though higher FBAP concentrations were seen at the latter.

*Acknowledgements.* The authors wish to thank the Large-scale Biosphere-Atmosphere (LBA) project group at the National Institute for Amazonian Research (INPA) in Manaus, for logistical support before and during the field deployment. The Brazil-UK Network for Investigation of Amazonian Atmospheric Composition and Impacts on Climate project was funded by the UK Natural Environment Research

Council (NERC; grant: NE/I030178/1). NERC also funded the Ph.D. studentship of E. Darbyshire. J. Brito was funded by Fundação de Amparo à Pesquisa do Estado de São Paulo (FAPESP; project 2013/25058-1).

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

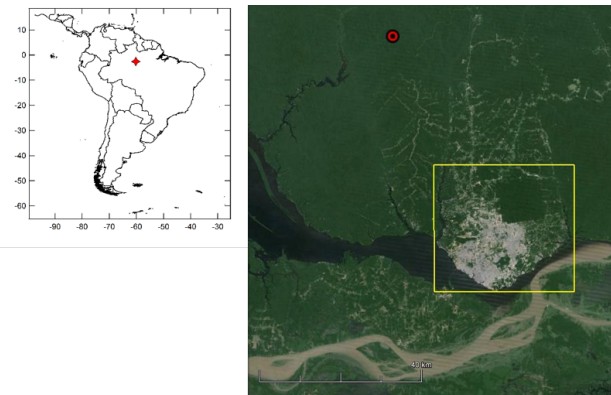

**Figure 1.** Location of the sampling site, shown as the red markers. The yellow rectangle represents the bounding box around Manaus used to flag air masses influenced by pollution from the city.

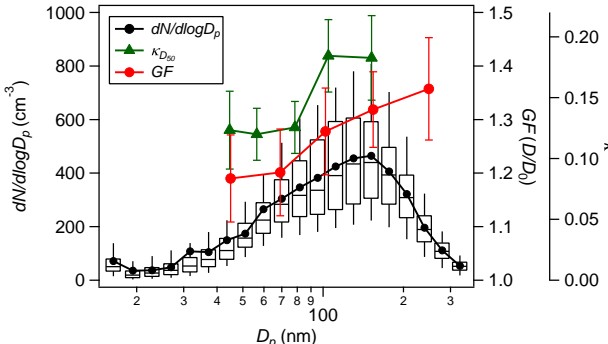

**Figure 2.** Particle number size distribution averaged over the entire measurement campaign. Box-and-whisker plots show the median, interquartile ranges and 5th and 95th percentiles, and lines and markers show mean $dN/dlogD_p$. Also shown are $\kappa$ derived from the $D_{50}$ from the CCNc, and growth factor from the HTDMA, both as a function of particle diameter. Error bars represent $\pm$ 1 standard deviation. Note that the HTDMA and CCNc / particle size data have been averaged over slightly different measurement periods, as shown in Figs. 4 and 7.

**Table 1.** Mean peak growth factors and derived $\kappa$ from HTDMA measurements for each dry diameter, along with $\pm$ standard deviation.

| $D_0$ (nm) | $GF$ | $\kappa$ |
|---|---|---|
| 45 | $1.19 \pm 0.08$ | $0.09 \pm 0.10$ |
| 69 | $1.20 \pm 0.08$ | $0.09 \pm 0.09$ |
| 102 | $1.28 \pm 0.08$ | $0.12 \pm 0.10$ |
| 154 | $1.32 \pm 0.07$ | $0.15 \pm 0.09$ |
| 249 | $1.36 \pm 0.10$ | $0.17 \pm 0.09$ |

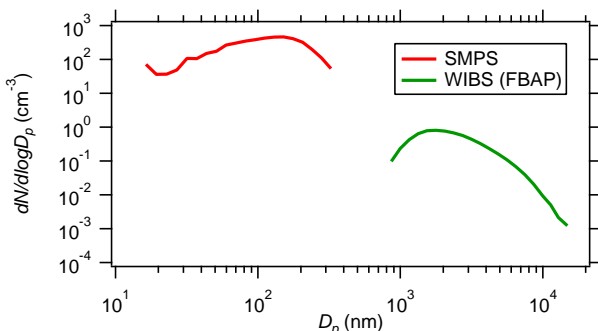

**Figure 3.** Combined log-log plot of total particle number size distribution (as measured with the SMPS) with FBAP number size distribution (from the WIBS). It should be noted that the SMPS measured the mobility diameters of particles in a dried sample, while the WIBS measured optical diameters at ambient humidity.

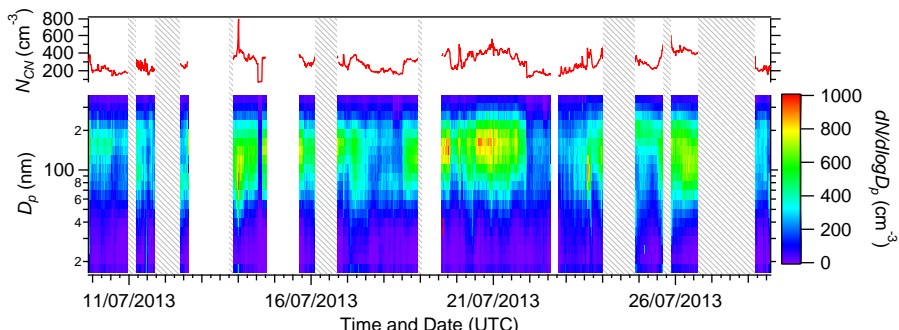

**Figure 4.** The time-series of total particle counts (integrated from size distributions; top panel) and particle number size distribution (bottom panel). Shaded areas represent pollution episodes removed from the data. Any other gaps are due to instrument down-time.

**Table 2.** Mean derived parameters from CCNc measurements for each set supersaturation ($SS$), along with $\pm$ standard deviation.

| $SS$ (%) | $D_{50}$ (nm) | $\kappa$ | $N_{CCN}$ (cm$^{-3}$) |
|---|---|---|---|
| 0.15 | $152 \pm 9.5$ | $0.18 \pm 0.03$ | $87 \pm 35$ |
| 0.26 | $105 \pm 5.5$ | $0.18 \pm 0.03$ | $161 \pm 60$ |
| 0.47 | $78 \pm 4.2$ | $0.13 \pm 0.02$ | $212 \pm 74$ |
| 0.80 | $56 \pm 3.0$ | $0.12 \pm 0.02$ | $248 \pm 82$ |
| 1.13 | $45 \pm 3.4$ | $0.12 \pm 0.03$ | $268 \pm 86$ |

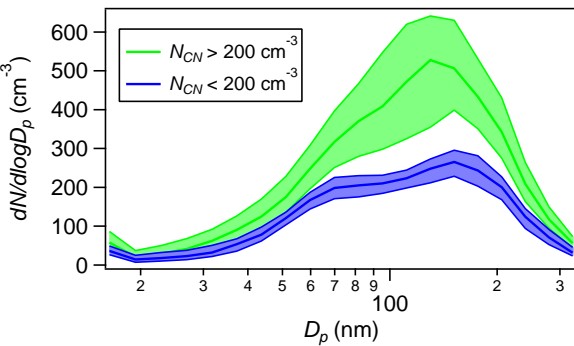

**Figure 5.** Median and interquartile ranges of particle number size distributions observed during high ($> 200$ cm$^{-3}$) and low ($< 200$ cm$^{-3}$) total particle number concentrations.

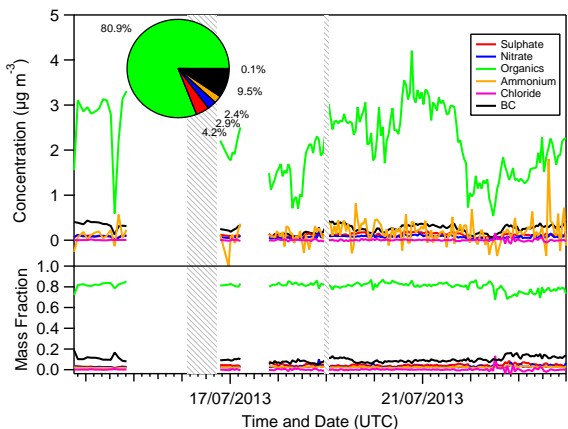

**Figure 6.** Submicron non-refractory aerosol composition from the ACSM measurements along with equivalent black carbon from the MAAP measurements: Concentration (top panel) and mass fraction (bottom panel). The pie chart shows the average proportions over the measurements. Shaded areas represent pollution episodes removed from the data. Any other gaps are due to instrument down-time.

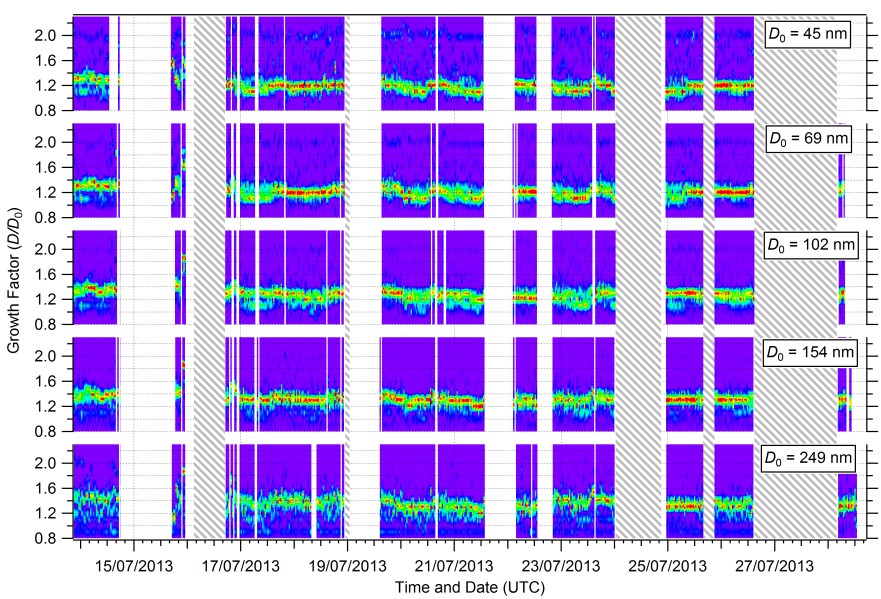

**Figure 7.** The time-series of normalised $RH$-corrected (to 90%) growth factor distributions derived from HTDMA measurements, for all 5 dry diameters. Shaded areas represent pollution episodes removed from the data. Any other gaps are due to instrument down-time and humidograms.

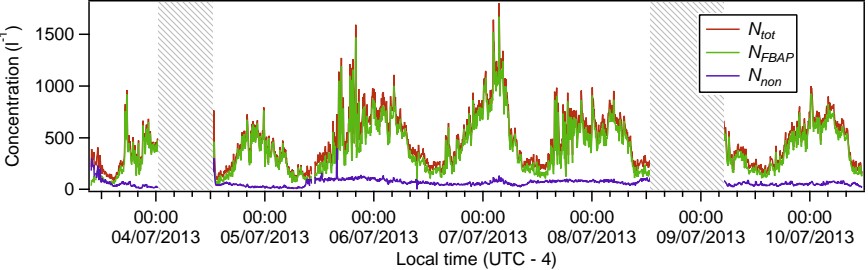

**Figure 8.** The time-series of total, FBAP and non-FBAP number concentrations as measured by the WIBS-3M. Shaded areas represent pollution episodes removed from the data. Any other gaps are due to instrument down-time.

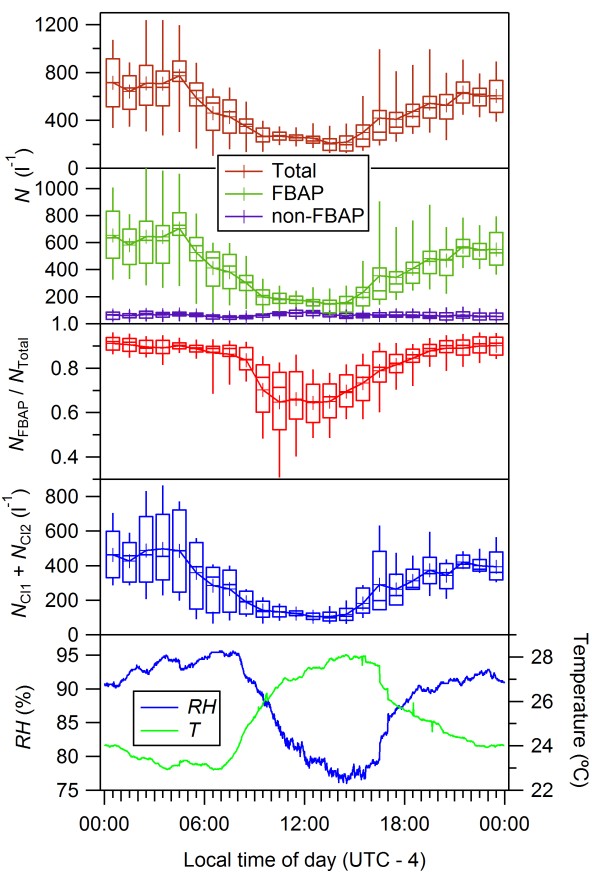

**Figure 9.** Diurnal variations in total, FBAP and non-FBAP number concentrations, as measured by the WIBS-3M, as well as the fraction of FBAP, and the combined number concentrations of clusters Cl1 and Cl2. Shown are the means (lines and markers), medians and inter-quartile ranges (boxes) and 5th and 95th percentiles (whiskers). Also shown at the bottom are the mean diurnal variations in temperature and $RH$, measured on the tower above the canopy, for the same period.

**Table 3.** Solutions to the Ward linkage cluster analysis, showing mean ($\pm$ 1 standard deviation) intensity in each fluorescence channel (FL1 - 3), optical particle diameter ($D_p$) and assymmetry factor ($A_f$). The intensities are referenced to the FT + 3 standard deviation threshold representing an intensity of zero, as discussed is section 2.5. Fluorescent intensities and assymmetry factor are in arbitrary units.

|  | Cl1 | Cl2 | Cl3 |
|---|---|---|---|
| FL1 (280 nm) | $1400 \pm 302$ | $478 \pm 386$ | $386 \pm 533$ |
| FL2 (280 nm) | $120 \pm 96$ | $33 \pm 47$ | $351 \pm 212$ |
| FL3 (370 nm) | $94 \pm 106$ | $47 \pm 73$ | $721 \pm 379$ |
| $D_p(\mu m)$ | $2.5 \pm 1.3$ | $1.9 \pm 1.0$ | $2.3 \pm 1.1$ |
| $A_f$ | $30.9 \pm 15.0$ | $30.2 \pm 15.7$ | $29.0 \pm 15.1$ |

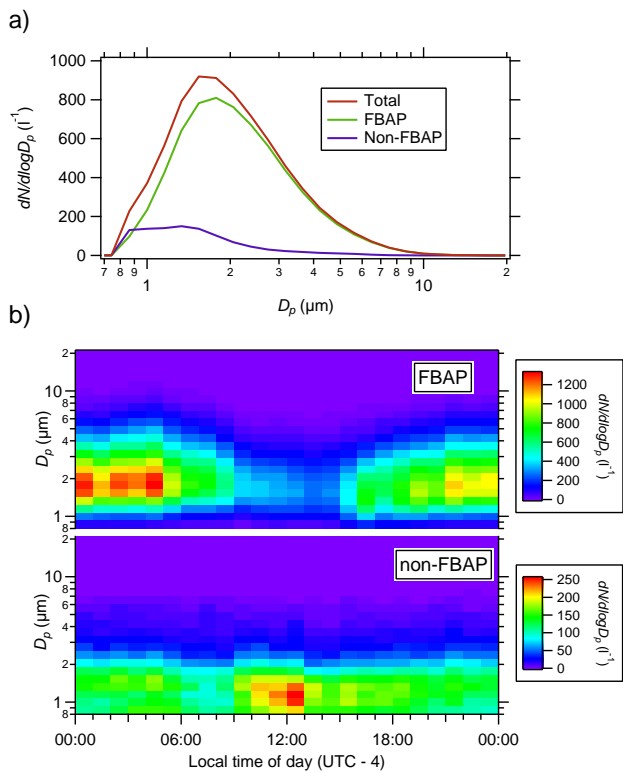

**Figure 10.** Particle number size distributions measured with the WIBS-3: a) mean size distributions for Total, FBAP and non-FBAP; and b) diurnal variation of size distribution for FBAP and non-FBAP (note that the colour scales are not the same).

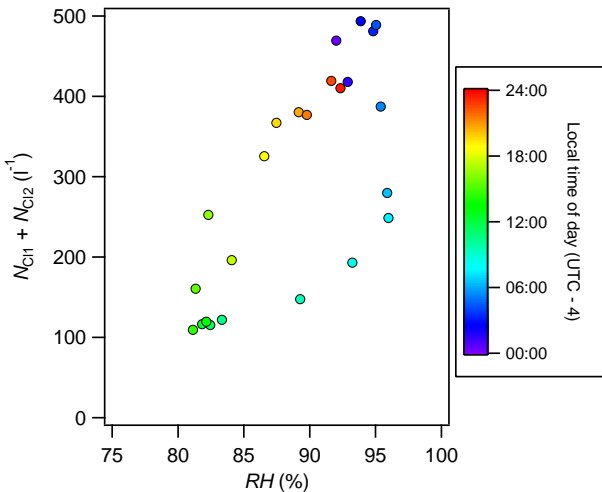

**Figure 11.** Diurnal means of total particle number concentrations in clusters 1 and 2, plotted against $RH$.

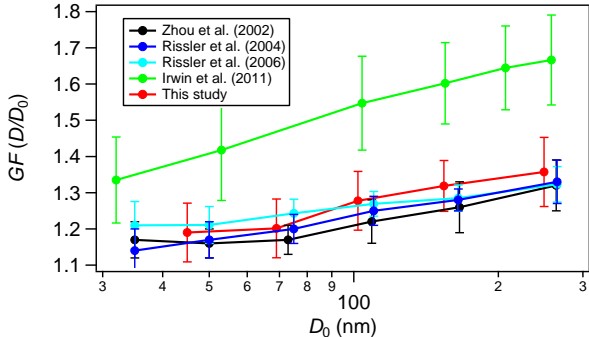

**Figure 12.** Mean growth factor for the dominant, less hygroscopic mode plotted against dry diameter, comparing this to previous studies in Amazonia and Borneo. The data from Rissler et al. (2004) and Rissler et al. (2006) represent "less" and "moderately hygroscopic" particles (respectively) during the wet season. The definitions differ slightly between the studies in terms of $GF$ range, but the modes represented here broadly fit into the "less hygroscopic" classification of Swietlicki et al. (2008). Error bars represent $\pm$ 1 standard deviation.

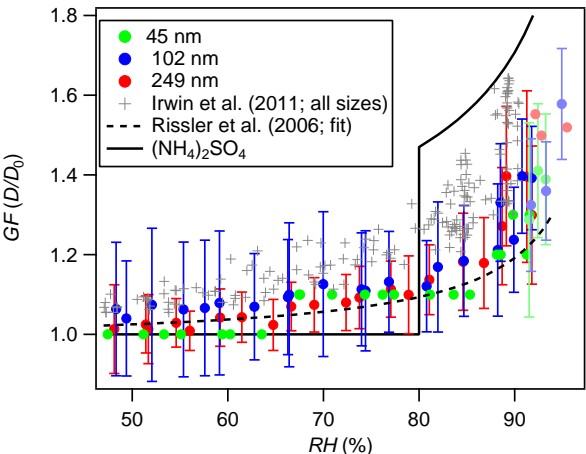

**Figure 13.** Humidogram (dependency of growth factor on $RH$), taken between 14:00 and 20:30 UTC on the 21st July. The fainter points at higher $RH$ were taken between 13:30 and 14:30 UTC on the 23rd July. The humidogram data from Irwin et al. (2011), and the humidogram fit from Rissler et al. (2006) are also shown, for comparison. The black line shows the modelled humidogram for ammonium sulphate (Topping et al., 2005) for reference.

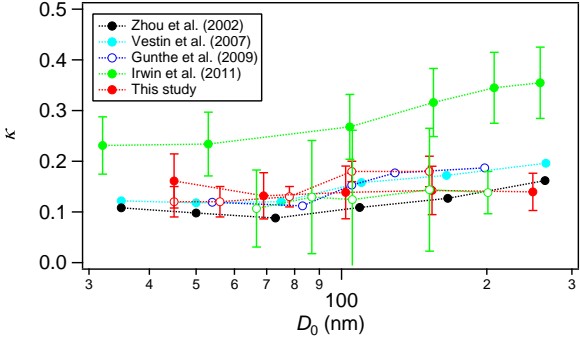

**Figure 14.** Comparing $\kappa$ as a function of diameter for this and previous studies in Amazonia and Borneo. Filled circles represent HTDMA derived values, while empty circles are CCNc derived values. Error bars represent $\pm$ 1 standard deviation, where this data is available. The values for Zhou et al. (2002) and Vestin et al. (2007) were calculated by Gunthe et al. (2009).