# Peer review of "Biogenic cloud nuclei in the Central Amazon during the transition from wet to dry season"

_Atmospheric Chemistry and Physics, 2015_

## Referee Comment (RC1) · Anonymous Referee #1 · 15 Feb 2016

Biogenic cloud nuclei in the Amazon

J.D. Whitehead, E. Darbyshire, J. Brito, H. M. J. Barbosa, I. Crawford, R. Stern, M. W. Gallagher, P. H. Kaye, J. D. Allan, H. Coe, P. Artaxo, and G. McFiggans

General comments:

This study describes measurements that seem to be of high quality in a very interesting region (the Amazon) during an interesting time that has previously not been character- ized (the transition period between wet and dry seasons). Given that these measure- ments fill an important gap, I recommend them ultimately for publication. However, I have many issues with the paper in its current state, and feel that it could be much im- proved. While I do not think there is a fatal flaw in the manuscript, there are instances where some additional basic analysis needs to be completed and sections that need further explanation or clarification. As it currently stands, the paper lacks enough of

this analysis, and is unclear enough in parts, that it should not be published.

Recommendation:

Before publication, major revisions need to be completed. I have tried to detail below those sections that either need further analysis or more detailed explanations.

Specific comments:

p3l15: The start date of the campaign is mentioned here. There should be some acknowledgement within this section that the WIBS measurements and all other measurements presented do not overlap. Perhaps this is completely unimportant insofar as the meteorology being similar over each week-long sampling period, but it should be acknowledged and discussed, if only briefly. If it is the case that these two weeks were very similar (meteorology, back-trajectories, etc.) and should be taken as descriptive of the same general time period, state so.

p3l18: Was this the same sampling location as AMAZE-08? Briefly state that this is the case if so.

p3l30: Attention to the details of the inlet seem to have been considered, but connect the dots for the readers: are there any significant particle losses? Assuming you have done those calculations, please state any relevant conclusions. This is especially important for the coarse mode. Assuming that there are no losses that need be accounted for, please state that you have done the calculations to verify so. Do not let the reader wonder or have to do the calculations themselves.

p6l30: You state "aerosol data were excluded if the pollution flag coincided with…" When I look at these plots, I see large gaps of data missing, e.g. maybe of 1/4 of the data in the Fig. 2 time series is absent. Should I conclude this is all pollution flagged? Or is some of it instrument down-time? It would be helpful if you could state in this paragraph what fraction of the data is removed due to pollution flagging. There is no data removed from the WIBS time series (fig. 6), which I assume means that there

were no pollution flags during this time? This relates somewhat to my earlier comment about the WIBS and all other data not over-lapping at all in time, and the question of how similar these two separate sampling periods actually are. It would be worth stating this explicitly, given how many gaps there seems to for the sub-micron instruments.

p5l26: You introduce the WIBS channels. Label them here as "FL1," "FL2," and "FL3."

p5l27: You are using FT+3sigma to define the FL threshold. Please provide a comment here on why are you not using the ambient threshold determination used by Perring et al (2015). It should be obvious to any instrument user, but it is worth explicitly stating that because the large majority of particles you are seeing are fluorescent, the ambient thresholding approach would not be appropriate. Also, please state what the actual threshold value of FT+3sigma is.

p5l30: "For a particle to be considered fluorescent. . ." Why are you using 3 sigma? There are numerous examples of different thresholds being used in WIBS studies (e.g. 2.5, 3, 4 sigma). Why is 3 picked? A citation should be provided here. I also recommend stating what the actual threshold value being applied is (i.e. the actual detector counts in the PMT), and not just what FT + 3sigma is. This is very important given that you report actual fluorescence intensity values in Table 3. Additionally, do these values of FT+3sigma stay constant over the measurement campaign? How often is FT mode run? More information on the data treatment here is needed. I would recommend conducting a sensitivity analysis on how different threshold value affect the fraction of particles determined to be fluorescent and the fluorescent particle concentrations. This would lend more meaning and context to the values reported in Table 3.

p5l34: What does it mean to "monitor instrument fluorescent channel efficiencies and baseline with time" using blue fPSLs?

p6l4: "Particles detected by this instrument" should be replaced with "Particles with fluorescent magnitudes about the threshold" or something similar (as the instrument "detects" both fluorescent and non-fluorescent particles via being an optical particle

counter).

p6l4: False-positive "FBAP" particles are a known issue in the WIBS. There are many WIBS studies (e.g. Toprak and Schnaiter 2013, Perring 2015 to name a few) and other single-particle fluorescence studies (e.g. Yong-Le Pan 2015) identifying non-biological fluorescent particles as interferences. There must be an acknowledgement within this section that molecules other than tryptophan and NADH fluoresce, some of which are not biological. Please also include any thinking or analysis you have done to identify the potential presence of false-positives in the WIBS. As it stands, without any discussion of interferences within the manuscript whatsoever, the following sentence should absolutely not be used: "Particles detected by this instrument...represent a lower limit of PBAP..."

p7l2: A general comment on size distributions: I would recommend adding a log-log version of Figure 1 (so have a Figure 1b perhaps) that shows size distributions over the entire size range, integrating the SMPS and WIBS data together. This would be a visual tool to very quickly convey how dominant the sub-micron mode is compared to the coarse mode in terms of particle number. Is it really true that there are no particles at e.g. 600nm (as Figure 8a indicates), or is this the WIBS detection efficiency going to zero? You state that the WIBS measures down to 500nm. Thus, the reasonable assumption from the reader is that there actually are no particles below 750nm, according to the WIBS. But how far does the accumulation mode (shown in Figure 1) tail extend to large diameters? Integrating these size distribution measurements would make all of this more clear.

p6l11: It seems that pollution episodes have been rigorously identified and removed. As the reader, though, I am wondering why these data were removed at all? Why not include that data, but identify it as potentially influenced by anthropogenic activities? This paragraph seems ideal to add another sentence or two as to explain further the rationale for why these episodes were removed.

p7l9: I find this discussion of Levoglucosan-as-tracer helpful, though am confused then why f60 is not used as a direct tool in section "2.5 Removal of pollution episodes." Was f60 only considered in the context of a campaign average? Simply because the campaign average is below a reported baseline, were there not episodes of BB influence as determined by the ACSM data directly, which has the ability to directly measure this? If not a graphical presentation of these results from the ACSM, there should at least be a mention of further analysis of BBOA composition that was done beyond looking at the campaign average of this tracer.

p7l24: Please include a paragraph on the presence or absence of PBAP markers from the ACSM data. This is an obvious omission given that at least one of the co-authors on this manuscript are among the very few that have used the AMS in an attempt to identify PBAP. Refer to Schneider et al 2011 ("Mass-spectrometric identification of primary biological particle markers and application to pristine submicron aerosol measurements in Amazonia").

p7l24: Another general comment on the Composition section: the utility of this paper, as I see it, is reporting what aerosol in the Amazon looks like during the transition between wet and dry seasons. Thus, solely reporting the organic, nitrate, and sulphate concentrations from the ACSM seems to be doing a disservice, and further analysis of the ACSM data could be included here. Was the aerosol oxidized? Were there any diurnal patterns in composition changes? How does the organic composition compare to the other studies citied? Should the conclusion drawn from the ACSM data be that there was basically no BBOA and similar organic concentrations compared to the other studies? (or, was there BBOA but it was flagged and removed?) More analysis and synthesis can be included here, given that the purpose of this paper is to give the community a baseline for this location in this season, and contrast it with the work that has been previously done. This seems to have been thoroughly done for the HTDMA data in section 3.5.1, but is absent for the aerosol composition data.

p9l10: Can you verify that the size-distribution in this figure is not 'fluorescence signal

limited?' It is possible, depending on the strength of the fluorescence from the material in these particles, that the signal strengths are on the same order as the threshold. If this were true and we assume an internally-mixed aerosol, there would thus be a particle size above which the average fluorescent signal would be greater than the threshold and below which the average fluorescent signal would be less than the threshold. This would make that size appear to be the true mode of the ensemble, but it would actually just be a reflection of the intrinsic fluorescent strength of the material within these particles. The 'true' diameter, so to speak, would be smaller than what it appears to be. Looking at a size-resolved average values of the FL signals for each FL channel would verify whether or not this data is in the regime. If not (and the signal strengths are sufficiently large relative to the applied threshold), this would add confidence to the reported mode diameter in Figure 8a. This is a general analysis issue for the fluorescent particle measurement community, and given that a paragraph of page 11 is devoted to comparing mode diameters between this and previous studies, I recommend this analysis.

p9l17: "Cl2 appears to be...somewhat less fluorescent." What exactly do you mean by saying 'less fluorescent?' Table 3 indicates the mode diameter of the cluster is 1.9um compared to 2.5um for Cl1. Would a 1.9um Cl1 particle have the same fluorescent intensity as a 2.5 um Cl2 particle?

p9l18: "Both clusters show similar fluorescent signatures to the clusters attributed to fungal spores by Crawford." How are the fluorescent signatures similar? In absolute intensity values? If that is the case, are the two instruments using the same detector gain settings, such that it would make sense to compare the intensities on an absolute scale? Or, are they similar in the relative strengths of channels Fl1-Fl2-Fl3? Even for relative differences between the channels, differences in gain settings would still be relevant in trying to compare this instrument's response with another. Further explanation and/or analysis on the spectral information collected by this WIBS should be provided to support the conclusion that these clusters represent fungal spores. There

are other WIBS studies that have identified WIBS signatures for fungal spores as well (see Healy 2012 in Atm Env; Perring, 2015 in JGR; Hernandez, 2016 in AMTD) that would be worth comparing your results to, perhaps here or in section 3.5.2.

p9l19: I find the following statement confusing: "These clusters (referring to Cl1 and Cl2) contribute approximately 70% to the total FBAP concentration, with no significant diurnal variation." Yet there is a very strong diurnal signal in FBAP, and Cl1+Cl2 makes up 70% of FBAP. Is there a typo here, or am I misunderstanding the phrase 'with no significant diurnal variation' in Cl1+Cl2?

p9l25: A general comment on this section: the comparison of the HTDMA data made during this study with other previous work done in the same region (or similar regions) seems well done. However, there has been plenty of work done previously on sub-micron aerosol composition in this region, and there is very little discussion of your ACSM data within the context of this previous work. Please add some content (perhaps a paragraph) in this section comparing your ACSM results to other measurements that have been made here in the Amazon.

p11l19: There are a number of studies not mentioned in this comparison section that the current manuscript would benefit from citing and discussing: -1. Poschl 2010: They attribute 80% of coarse-mode particles as primary biological particles. While those measurements were done with SEM, they seem to align with these results and should be mentioned. -2. Please also include in this paragraph how your results compare to PBAP modeling work that covers this region (e.g. Spracklen and Heald 2014). -3. A recent study on fungal spore measurements in the coarse mode, "Significant influence of fungi on coarse carbonaceous and potassium aerosols in a tropical rainforest." by Zhang and co-workers. They estimate fungal spore concentrations in a similar environment. There may be more studies. As this section is meant to compare your results to what has come before, a more thorough review of the literature should be done, and should not just be limited to aerosol fluorescence measurements as there are other ways of determining concentrations of airborne fungal spores.

p12l10: Similar to an earlier comment, it is not clear to me if there was no data recorded of biomass-burning influenced air, or if there was the influence of biomass burning but those data were flagged and removed. You write here "…the results here may reflect the transition between the two seasons, with periods consistent with each at different times (but without any influence from biomass burning)." The confusion arises because I am left wondering if the air sampled during this period is similar when you discount biomass burning influence, or if the air sampled here is similar partially because there is no biomass burning influence.

Technical corrections:

p6l27: This does not need to be a new paragraph.

Figure 1: Change "Particle number size distribution for the experiment" to "Particle number size distribution averaged over the entire measurement campaign" or something similar. Also, there are kappa and GF data here as well, which should also be mentioned in the caption.

Figure 2: Change caption to "The time-series of total particle counts (top panel) and particle number size distribution (bottom panel)." The order of what you list should go top to bottom, and with the multiple panel figures explicitly naming what is where reduces any possible confusion.

Figure 4: Can you make use of the entire range of the ROYGBIV colorscale? Almost all of the data is blue-ish/green, making use of the rest of the scale would make the data more visible here. Also given that this figure comes after a previous figure with many gaps, I would move the gaps statement ("Gaps are largely due…") up to Figure 2 or include this statement in each caption.

Figure 5: I assume this is the case, but is the pie-chart for the average of all the data shown here? State this briefly in the figure caption.

Figure 9: "Mean growth factor for the dominant less hygroscopic mode" should be

"Mean growth factor for the dominant, less-hygroscopic mode." Also typo with "agains."

Table 3: What are the units here? E.g. there should be units next to Cl1, Cl2, etc. What are the units of Asymmetry factor? (else a definition of what "Af" actually is should be provided somewhere in the text)

[Figure]

---

## Referee Comment (RC2) · Anonymous Referee #2 · 16 Feb 2016

The paper 'Biogenic cloud nuclei in the Amazon' presented by Whitehead et al. contains a detailed compilation of different measurements during a 3-weeks intensive in the transition period between wet and dry season at a remote research station in the Amazon. The authors focused on different measurements of micro-physical, chemical and hygroscopic properties of the sub-micron aerosol particle population as well as the fluorescence of super-micron particles - a thoroughly interesting, comprehensive and significant data set. The collected data and shown results are relevant to the scientific community and contribute to a deeper understanding of the significance of (biogenic) aerosol particles for cloud properties and the formation of (mixed-phase) precipitation and hence the hydrological cycle in the Amazon.

The subject matter is clearly in the area of ACP. Nevertheless, I think several aspects concerning the data analysis and further technical issues need to be revisited carefully before the manuscript can be accepted for publication in ACP. Please find my major

comments below.

**General Comments:**

The manuscript shows an interesting but brief compilation of individual data sets, which are finally compared to previous studies. Since the whole data set comprises (as stated by the authors) a large variability e.g., for the total particle number concentration (100 - 800 cm-3, cf. Fig. 2), shape of the particle number size distribution (cf. Fig. 1), organic mass contribution measured by the ACSM (0.5 - 4 $\mu$g m-3, cf. Fig. 5), one would expect to find similar variability in GF or kappa. Nevertheless, GF and kappa are mainly discussed in terms of campaign averages and the applied color scale in Fig. 4 makes it hard to identify variability. Interestingly, the time series of GF does show clear episodes of stable conditions (cf. July 22th) versus episodes with higher variability (cf. July 23rd). Furthermore, during a short event on July 15th GF shows extraordinary high values (> 1.6), which is not discussed in the manuscript.

I suggest to carefully revisit the results section towards a more systematic and comprehensive analysis and discussion combining information from different measurements (particle number size distribution, total particle number concentration, hygroscopicity and chemical information).

The authors apply a hierarchical cluster analysis to the WIBS data, which is certainly a powerful technique to identify PBAB meta-classes. However, there is significant information missing about the input to the analysis and the corresponding discussion. This paragraph is not clearly outlined making it hard to follow the argumentation.

Finally, the title is very unspecific and does not clearly reflect the content of the paper.

I summarize more specific comments below.
**Specific comments:**

Section 2.1:

- first paragraph: The authors compare rainfall, temperature and humidity during their measurement period with AMAZE-08. Please specify the statement 'cooler and more humid'.

- second paragraph: This paragraph deals with detailed information on the location of the measurement site. Please consider to add a map. This would also be helpful for the discussion concerning the removal of pollution episodes.

Section 2.5:

- The authors describe how they flag and remove pollution episodes from the entire data set. Last sentence: 'Approximately 28% of the HTDMA and CCNc data were removed in this way, with 5% of the data being flagged as possibly impacted by biomass burning and most of the rest due to the Manaus urban plume.'

- Why are only HTDMA and CCNc data removed? Additionally, data gaps in the shown figures have to be specified.

- I further suggest to consider to show a figure containing all geographical information including the mentioned Manaus bounding box.

Section 3.2:

- In section 2.5 the authors already introduce a 'cleaning procedure' to exclude pollution episodes. Does $f_{60}$ show any correlation with the detected pollution events?

- p. 7, ll. 21: 'The mean $f_{60}$ at TT34 in July 2013 was 0.19% $\pm$ 0.07%. This is well below 0.3%, which is considered to be the upper limit for background air masses not affected by biomass burning' Have the ACSM data been filtered? Is the mean value calculated after removing pollution events?

Section 3.4:

- p. 8, l. 18: 'mean total particle number concentration of FBAP ..' Do you mean the mean FBAP or the mean total particle number concentration?

- p. 8, l. 31: 'The observed night-time peak in FBAP number concentrations in fig. 7 is consistent with nocturnal sporulation driven by increasing RH' Where did you measure T and RH? Are the measurements collocated (below or above canopy) or part of the regular measurements at the research tower (if so, at which height)?

- p. 9, l. 8: '... FBAP clearly dominates the particle number concentrations for $D_p > 1$ $\mu$m, however non-FBAP concentrations are higher for submicron particles': How robust is the characterization of the WIBS instrument? I wonder if this statement might be influenced by a decrease in sensitivity of the fluorescence signal. According to Crawford et al. (2015), the WIBS-4 has a 50% detection diameter at 0.8 $\mu$m. Please specify the 50% detection diameter of your instrument.

- p. 9, ll. 13: The authors apply a cluster analysis to the WIBS data without providing details on the data preparation and the precise input. According to the cited paper by Crawford et al. (2015), several steps are involved to filter the data before clustering. Did the authors apply exactly the same criteria? Even if so it is worth mentioning those criteria and the corresponding rejection rate in this manuscript.

- p. 9, ll. 15: It is hard to follow the argumentation concerning the cluster analysis: 'Cl1 has previously been attributed to fungal spores (Crawford et al., 2014) based

on comparison with other sampling techniques and the diurnal emission pattern (see fig. 7) with higher concentrations observed overnight' Was Cl1 attributed to fungal spores based on the observed diurnal cycle (in this publication) or on the mean values (of FL1-3, AF, size) of the corresponding cluster in Crawford et al., 2014?

- p. 9, ll. 20: 'The statistical parameters of each cluster are shown in table 3 for comparison. Together, these clusters contribute approximately 70% to the total fluorescent particle concentration, with no significant diurnal variation in this figure, suggesting that FBAP were dominated by fungal spores during this study.' Why does the hierarchical cluster analysis cluster only 70% of the data? Why is there no significant diurnal variation? And why does it in this case lead to the stated conclusion?

Section 3.5.1:

- p. 10, l. 28: 'The HTDMA derived $\kappa$ from the Borneo experiment shows more hygroscopic aerosol than in Amazonia, as discussed above, however the CCNc derived values are more in line with those in Amazonia. This discrepancy has been noted previously and possible reasons for it discussed by Irwin et al. (2011) and Whitehead et al. (2014).' It would be interesting to discuss the findings of the mentioned papers in the context of the here observed discrepancy.

Section 3.5.2:

- p. 11, l. 7: 'The median number concentration of FPAB observed below the canopy in this study was 372 l−1'. Unprecise – which study do you mean, Gabey et al. (2010) or this study?

- Concerning the observed discrepancies with Huffmann et al. (2012), the authors discuss instrumental issues, mixing effects related to strong vertical gradients

and pbl development. I suggest to add a discussion about possible effects of wet deposition, since the measurements of Huffmann et al. (2012) were performed during the wet season.

- p. 11, l. 28: 'Diurnal variations between this study and that of Huffman et al. (2012) were similar, however Gabey et al. (2010) reported an additional increase in the afternoon in Borneo'. Unprecise – which measurement parameter increases?

**Technical issues:**

Please reference all your physical variables in the text and/or figure captions.

Please do not use abbreviations like 'don't' (e.g., p. 11, l. 32).

Figure captions miss significant information:

Fig. 1:

- information on the derived GF and kappa is missing

- HTDMA, CCNc data comprise different measurement periods. Please specify that in the figure. Are these data averaged over the same time period?

Fig. 2:

- $N_{CN}$ - is this measured by the CPC or integrated from the size-resolved measurements?

- please specify the data gaps

Fig. 4:

- please specify the data gaps

- all other figures use GF($D/D_0$) instead of 'Growth Factor $D/D_0$'

Fig. 5:

- please specify the data gaps

- The unit is probably $\mu$g/m3

- What is the collection efficiency for the ACSM data?

Fig. 6:

- $N_{tot}$ refers to the size range of the WIBS, make sure that there is no confusion with the term 'total counts' in Fig. 2

Fig. 7:

- $N_{tot}$ refers to the size range of the WIBS, make sure that there is no confusion with the term 'total counts' in Fig. 2

- please add information about the sensor height and position for T and RH

- $^{\circ}C$

Fig. 8 a & b:

- you use Dp instead of $D_p$

- unit of dN/dlog dp is wrong

Fig. 9: 'Irwin et al., (2011)'

References:

- page 16, line 15: lower case initials: 'Wiedensohler, Arana'

- page 17, line 3: full name instead of initials: 'Anna Stefaniak'

---

## Referee Comment (RC3) · Anonymous Referee #3 · 7 Mar 2016

This manuscript reports the results of aerosol measurements taken place in the Amazon basin during the wet-to-dry transition period. The measurements include particle size distributions, hygroscopicity, and fluorescent biological aerosol particle concentrations, and are compared to the previous measurements. The results are important and interesting, especially since there are few previous studies in that environment. However, it is not clear to me why the authors choose to remove pollution episodes from this dataset and how this "clean" dataset provides a "unique contrast (page 2, line 10) to the wet-season data?" In fact, the observed particle total number concentrations and hygroscopicity as well as chemical composition are quite similar to those observed during the wet season. The WIBS-3 results are different but also largely because the measurements were done within the canopy. To me, the removed data are really the key feature of the transition period, meaning influences but not as strong as the dry season. It is important to add that analysis as a contrast. The authors should also pay attention to the manuscript preparation

guidelines for authors provided by the journal (http://www.atmospheric-chemistry-and-physics.net/for_authors/manuscript_preparation.html). I recommend this manuscript be published after the following comments are addressed.

Specific comments:

(1) A 5-paragraph abstract seems unnecessary for this paper. Some of the details may be removed and the key points need to be summarized more concisely.

(2) Page 3, line 19-20: Please provide the relative humidity and temperature for both campaigns.

(3) Page 4, line 30; Page 5, line 32: Do you mean "polystyrene latex spheres (PSL)" for both cases? What sizes have you used for the calibration? Do the uncertainties for growth factor derived from HTDMA vary by D0? What do you mean "blue fluorescent latex spheres"? Please clarify. Also, since different kinds of diameters are described in the paper, the authors should specify the diameter type in the text and figures.

(4) Page 6, line 4-5: Do you mean "some of the PBAP are detected by WIBS"? Please clarify and give examples.

(5) Section 2.5: It is not clear to me which flag was applied to which dataset and whether if the flag was properly set. The authors should provide clear information about the data processing and have consistency among datasets.

First of all, Figures 2, 4, and 5 look like having different gaps (lack of clear description in the graphs and figure captions about the gaps).

Second, the back trajectories at all altitudes from 0 to 4000 m.a.s.l were used for the identification of pollution episodes (page 6, line 17). However, most sampling was taken at 39 m (10 m above canopy) and WIBS was operated on the ground level.

Third, it was said that data sampled for local wind direction of 270ËŽ-340ËŽ were flagged as potential generator contamination (line 27). But in line 31-32, the authors

said that 5% of the removed data were potential biomass burning and the rest were Manaus plume. Then, which part is due to generator contamination?

Finally, in line 29-30, significant increases in black carbon concentration and particle number concentration were used as the second criteria of data removal. The question is "are there periods with such significant increases but not flagged by the back trajectories passed over Manaus, fire zone, or by wind direction for generator plumes?" If so, when and why? If not, the former (increases) is enough for identifying the pollution episodes.

(6) Section 3.1: Both paragraphs said that the observed particle number size distributions are similar to the ones measured in the dry season (i.e., effected by biomass burning). However, the data are supposed to represent background conditions because of the removal of pollution episodes.

(7) Page 7, line 21-23: The author should clarify that the ACSM data (Fig. 5) do not cover the entire measurement period (Figs 2 and 4). "in July 2013" is inaccurate. Have the excluded periods flagged by biomass burning shown elevated f60?

(8) Page 7, line 28: What are the definitions of hydrophobic, less or more hygroscopic mode (page 10, line 2) in terms of growth factor? Are their definition consistent in literatures (e.g., for the comparisons done in page 10, line 1-12?

(9) Page 7, line 29-30: What does the "local anthropogenic influence" stand for? What is "this distribution (i.e., . . .)"?

(10) Page 8, line 1-5: Increased growth factor with particle dry diameter can be explained by many possibilities (it doesn't have to be greater sulfate contribution at larger diameter; organic material at different diameter may different as well). Without careful analysis, I think it is hard to demonstrate that the observations here reflect similar size-resolved chemical information to the previous studies. And the particle number size distributions observed in this study are indeed different from what was observed

in previous wet-season studies as described in Sect. 3.1.

(11) Page 9, line 15-24: The analysis here is confusing and needs clarifications.

It was said first that C11 is attributed to fungal spores and C12 remain unclassified. Then why "both clusters show similar fluorescent signatures to the clusters attributed to fungal spores"? Aren't all the three classes distinct in fluorescent signatures (line 15)?

Second, in line 21, it was said that "these clusters ..., with no significant diurnal variation in this figure, suggesting that FBAP were dominated by fungal spore during this study." Does "these" mean C11+C12 or C11+C12+C13? Don't C11 and C12 show nighttime increase in Fig. 7? Finally, if C13's concentration is low, what about the residuals in the cluster analysis (meaning Fig. 7 showed a difference of hundreds in number concentration between FBAP and C11+C12)? What does the "insufficient data" mean in line 24?

(12) Page 10, line 9 and line 12: What does "strong diurnal cycles" mean? Daytime peak? Please clarify.

(13) Page 10, line 31-32: What about the removed data? Do those data show very different results compared to the "clean" conditions? Also it is important to explain why the particle concentrations and hygroscopic properties are similar to those during the wet season but the particle size distributions are similar to those observed in the dry season (my comment #6, Sect. 3.1).

(14) Page 11, line 11-12: What kind of meteorological conditions? Need a reference or example to support this hypothesis. Also, what are "other locations"? Please specify.

Technical remarks: Page 3, line 18-19: Revise "the AMAZE-08 campaign saw 370 mm fall" and move the reference to the end. Page 3, line 27: Revise "local time was UTC – 4 hours". Page 3, line 31: "RH" has not been defined yet. Page 4, line 21 and 25: Properly revise "dry sizes" since the DMA selects a band of the electric mobility not just

one size. Page 4, line 28-29: "a bubble flowmeter" is an improper description. Also, shouldn't be "Gillibrator-2"? Page 5, line 23-26: What is NADH? What do you mean "3 fluorescence channels"? Page 6, line 4: Add "as" after "termed" and revise the later part of the sentence. Page 7, line 3 and later text: "fig. " should be "Fig. ". Page 7, line 8: "particle counts" should be "particle number concentrations". Figure 5. Remove frame. Figure 5 appeared earlier than Fig. 4. Page 7, line 27: Should be "in the range of 1.2 to 1.4" (the word "of" is missing). Page 8, line 8-9: Check the grammar for " at larger diameters $\kappa \approx$ . . . and $\kappa \approx 0.18$ around the accumulation mode. " SI units should be used, and units in the denominator should be formatted with negative exponents.

---

## Author Comment (AC1) · 10 May 2016

**Response to Anonymous Referee #1**

General comments:

This study describes measurements that seem to be of high quality in a very interesting region (the Amazon) during an interesting time that has previously not been characterized (the transition period between wet and dry seasons). Given that these measurements fill an important gap, I recommend them ultimately for publication. However, I have many issues with the paper in its current state, and feel that it could be much improved. While I do not think there is a fatal flaw in the manuscript, there are instances where some additional basic analysis needs to be completed and sections that need further explanation or clarification. As it currently stands, the paper lacks enough of this analysis, and is unclear enough in parts, that it should not be published.

We thank the referee for taking the time to thoroughly review our manuscript. We have addressed each of the comments and recommendations below, and will revise the manuscript accordingly.

Recommendation:

Before publication, major revisions need to be completed. I have tried to detail below those sections that either need further analysis or more detailed explanations.

Specific comments:

p3l15: The start date of the campaign is mentioned here. There should be some acknowledgement within this section that the WIBS measurements and all other measurements presented do not overlap. Perhaps this is completely unimportant insofar as the meteorology being similar over each week-long sampling period, but it should be acknowledged and discussed, if only briefly. If it is the case that these two weeks were very similar (meteorology, back-trajectories, etc.) and should be taken as descriptive of the same general time period, state so.

The referee is correct to note that the sample periods do not overlap for these instruments. In terms of meteorology, the conditions were very similar of the whole measurement period, as the referee suggests. We will include a short paragraph at the end of this section discussing this.

p3l18: Was this the same sampling location as AMAZE-08? Briefly state that this is the case if so.

It was, and we will clarify this here.

p3l30: Attention to the details of the inlet seem to have been considered, but connect the dots for the readers: are there any significant particle losses? Assuming you have done those calculations, please state any relevant conclusions. This is especially important for the coarse mode. Assuming that there are no losses that need be accounted for, please state that you have done the calculations to verify so. Do not let the reader wonder or have to do the calculations themselves.

For the range of flows rates during BUNIAACIC the transmission range has previously been calculated from 4nm to 7µm (Martin et al., 2010). As mentioned in the previous reply, the experiment was conducted at the same sampling location as AMAZE-08 and we will refer to this characterisation in the manuscript.

p6l30: You state "aerosol data were excluded if the pollution flag coincided with. . ." When I look at these plots, I see large gaps of data missing, e.g. maybe of 1/4 of the data in the Fig. 2 time series is absent. Should I conclude this is all pollution flagged? Or is some of it instrument down-time? It would be helpful if you could state in this paragraph what fraction of the data is removed due to pollution flagging. There is no data removed from the WIBS time series (fig. 6), which I assume means that there were no pollution flags during this time? This relates somewhat to my earlier comment about the WIBS and all other data not over-lapping at all in time, and the question of how

similar these two separate sampling periods actually are. It would be worth stating this explicitly, given how many gaps there seems to for the sub-micron instruments.

We will signify in the time-series figures (by shaded area, or similar) the periods removed due to pollution flags. Regarding the WIBS data, there was a mistake in that two pollution episodes were not removed from the time-series in figure 6. This makes no difference to the results or conclusions, and we will modify the figure with removed data specified in the same way as the other figures.

p5l26: You introduce the WIBS channels. Label them here as "FL1," "FL2," and "FL3."

We will add these labels in the appropriate places in the revised manuscript.

p5l27: You are using FT+3sigma to define the FL threshold. Please provide a comment here on why are you not using the ambient threshold determination used by Perring et al (2015). It should be obvious to any instrument user, but it is worth explicitly stating that because the large majority of particles you are seeing are fluorescent, the ambient thresholding approach would not be appropriate. Also, please state what the actual threshold value of FT+3sigma is.

The fluorescence threshold value of FT mean + 3 standard deviations was agreed upon by the WIBS community as the standard for determining particle fluorescence at the 2014 WIBS user group meeting (Boulder, CO, USA) and this value is used in other publications using the same instrument used here (Robinson et al, 2013, Crawford et al, 2014, Crawford et al, 2015, Crawford et al, 2016) so we use this value for consistency.

The method employed in Perring et al (2015) was used to constrain periods where the baseline was unusually variable, most likely due to the presence small fluorescent particles that were below the instrument's size detection limit or fluorescent vapours (e.g., acetone) which would increase the fluorescent background of the optical chamber. This method is unsuitable at the sampling site for the reasons suggested by the referee, and we will state this in the revised manuscript.

As requested, the FT mean +3σ thresholds were: FL1 112.4 ± 3.9, FL2 284.6 ± 7.8, FL3 164.6 ± 5.7.

p5l30: "For a particle to be considered fluorescent. . ." Why are you using 3 sigma? There are numerous examples of different thresholds being used in WIBS studies (e.g. 2.5, 3, 4 sigma). Why is 3 picked? A citation should be provided here. I also recommend stating what the actual threshold value being applied is (i.e. the actual detector counts in the PMT), and not just what FT + 3sigma is. This is very important given that you report actual fluorescence intensity values in Table 3. Additionally, do these values of FT+3sigma stay constant over the measurement campaign? How often is FT mode run? More information on the data treatment here is needed. I would recommend conducting a sensitivity analysis on how different threshold value affect the fraction of particles determined to be fluorescent and the fluorescent particle concentrations. This would lend more meaning and context to the values reported in Table 3.

The rationale for using a threshold value of FT mean + 3 standard deviations is discussed in response to the previous comment and citations to the relevant publications (Robinson et al, 2013, Crawford et al, 2014, Crawford et al, 2015, Crawford et al, 2016) will be provided in the revised manuscript.

During data processing the threshold value for each channel is subtracted from the single particle fluorescence data and the value is clipped at 0 with all values greater than 0 being considered significantly fluorescent compared to the instrument baseline. Fluorescence measurements below the threshold (i.e. less than 0 after threshold subtraction) are not considered physically meaningful and are clipped at 0. This is described in Crawford et al (2015) and we will include a short description of the processing method in this section. As such the fluorescent intensity values reported in Table 3 are relative to the applied threshold and not the absolute detector intensity.

The threshold remains consistent where 58 FT samples were made over the course of the campaign (see earlier response).

p5l34: What does it mean to "monitor instrument fluorescent channel efficiencies and baseline with time" using blue fPSLs?

This statement was misleading. In fact, the fPSLs were just used at the start of the measurements to check that the instrument was working properly. We will reword this accordingly.

p6l4: "Particles detected by this instrument" should be replaced with "Particles with fluorescent magnitudes about the threshold" or something similar (as the instrument "detects" both fluorescent and non-fluorescent particles via being an optical particle counter).

Correct. We will change the wording as the referee suggests.

p6l4: False-positive "FBAP" particles are a known issue in the WIBS. There are many WIBS studies (e.g. Toprak and Schnaiter 2013, Perring 2015 to name a few) and other single-particle fluorescence studies (e.g. Yong-Le Pan 2015) identifying nonbiological fluorescent particles as interferences. There must be an acknowledgement within this section that molecules other than tryptophan and NADH fluoresce, some of which are not biological. Please also include any thinking or analysis you have done to identify the potential presence of false-positives in the WIBS. As it stands, without any discussion of interferences within the manuscript whatsoever, the following sentence should absolutely not be used: "Particles detected by this instrument. . .represent a lower limit of PBAP. . ."

We will include a discussion of fluorescent interferents in the revised manuscript. Generally the identified interferents are smaller than the detection limit of the WIBS; polycyclic aromatic hydrocarbons (PAH) such as naphthalene have been shown to fluoresce in Fl1 (Pöhlker et al., 2012). Soot containing such interferent PAH's have also been investigated; Propane flame soot was generated at a C/O ratio of 0.5 and coagulated in a small aerosol processing chamber to detectable sizes ($D_p$ > 0.8 μm) prior to sampling with a WIBS-4 where it was found that 0.2% of the soot population would fluoresce in Fl1 (Toprak and Schnaiter 2013). We would not expect to observe significant concentration of PAH's or soot outside of the pollution events at such a remote site so their contribution to the observed fluorescent concentration should be negligible.

Mineral dusts contain a small subset of fluorescent aerosol within their population (~10%), and given their ubiquitous nature may present a significant source of interferents to the UV-LIF method (Toprak and Schnaiter 2013), however their observed fluorescent intensity is considerably weaker than is observed for biofluorophores (Pöhlker et al., 2012) and if they were present in any significant concentration they would likely form their own cluster as was demonstrated in Crawford et al. (2016). We will add a brief discussion on this in the revised manuscript.

It is also worth adding that the technique measures "biological containing particles", which may include fluorescent material attached to non-biological particles. We will include a brief explanation of this in the revised manuscript.

p7l2: A general comment on size distributions: I would recommend adding a log-log version of Figure 1 (so have a Figure 1b perhaps) that shows size distributions over the entire size range, integrating the SMPS and WIBS data together. This would be a visual tool to very quickly convey how dominant the sub-micron mode is compared to the coarse mode in terms of particle number. Is it really true that there are no particles at e.g. 600nm (as Figure 8a indicates), or is this the WIBS detection efficiency going to zero? You state that the WIBS measures down to 500nm. Thus, the reasonable assumption from the reader is that there actually are no particles below 750nm, according to the WIBS. But how far does the accumulation mode (shown in Figure 1) tail extend to large diameters? Integrating these size distribution measurements would make all of this more clear.

We have rejected particles smaller than 800 nm from the analysis due to low collection efficiency. We will clarify this in the revised manuscript. Figure 1 shows the SMPS and WIBS size distributions together in a log-log plot. Unfortunately, there is a considerable gap between the size ranges of the two instruments, plus there are issues with trying to combine the dry mobility diameters from the SMPS with the wet optical diameters from the WIBS. The figure doesn't show how far the accumulation mode tail extends or how much it contributes to the coarse mode. We feel it therefore does not add anything to the paper, and have decided not to include it.

[Figure]

Figure 1. Combined log-log plot of total particle number size distribution (as measured with the SMPS) with FBAP number size distribution (from the WIBS).

p6l11: It seems that pollution episodes have been rigorously identified and removed. As the reader, though, I am wondering why these data were removed at all? Why not include that data, but identify it as potentially influenced by anthropogenic activities? This paragraph seems ideal to add another sentence or two as to explain further the rationale for why these episodes were removed.

The focus of this paper is on the natural (biogenic) aerosol at this time of year to compare with the wet season. We will clarify the scope of the paper in the introduction of the revised manuscript to explain why the pollution events were removed.

p719: I find this discussion of Levoglucosan-as-tracer helpful, though am confused then why f60 is not used as a direct tool in section "2.5 Removal of pollution episodes." Was f60 only considered in the context of a campaign average? Simply because the campaign average is below a reported baseline, were there not episodes of BB influence as determined by the ACSM data directly, which has the ability to directly measure this? If not a graphical presentation of these results from the ACSM, there should at least be a mention of further analysis of BBOA composition that was done beyond looking at the campaign average of this tracer.

Previous studies in the Amazon have observed that a large fraction of the biomass-burning related organic aerosols do not present a significant f60 signal, due to long-range transport (Brito et al., 2014). As such, applying a f60 threshold would remove only fresh fires and not biomass burning emissions. We accept that the text isn't at all clear on this, and we will revise it to clarify this point. The statement in the current text simply says that the relatively low f60 confirms for us that there was no sign of local BB influence during measurements.

p7l24: Please include a paragraph on the presence or absence of PBAP markers from the ACSM data. This is an obvious omission given that at least one of the co-authors on this manuscript are among the very few that have used the AMS in an attempt to identify PBAP. Refer to Schneider et al 2011

("Mass-spectrometric identification of primary biological particle markers and application to pristine submicron aerosol measurements in Amazonia").

The following text has been added to the manuscript:

"Previous studies have successfully identified FPAB markers on ambient aerosol in the Amazon using an aerosol mass spectrometer (Schneider et al., 2011), a method which relies strongly on the high-resolution capabilities of the instrument used at the time. Given the unity mass resolution of the ACSM, similar methodology has not been applied here.

p7l24: Another general comment on the Composition section: the utility of this paper, as I see it, is reporting what aerosol in the Amazon looks like during the transition between wet and dry seasons. Thus, solely reporting the organic, nitrate, and sulphate concentrations from the ACSM seems to be doing a disservice, and further analysis of the ACSM data could be included here. Was the aerosol oxidized? Were there any diurnal patterns in composition changes? How does the organic composition compare to the other studies citied? Should the conclusion drawn from the ACSM data be that there was basically no BBOA and similar organic concentrations compared to the other studies? (or, was there BBOA but it was flagged and removed?) More analysis and synthesis can be included here, given that the purpose of this paper is to give the community a baseline for this location in this season, and contrast it with the work that has been previously done. This seems to have been thoroughly done for the HTDMA data in section 3.5.1, but is absent for the aerosol composition data.

The reviewer is correct as there is a lot to explore from aerosol mass spectrometry measurements during BUNIAACIC campaign. The authors see fit that such detailed description would suit better a separated manuscript, which is currently under preparation.

p9l10: Can you verify that the size-distribution in this figure is not 'fluorescence signal limited?' It is possible, depending on the strength of the fluorescence from the material in these particles, that the signal strengths are on the same order as the threshold. If this were true and we assume an internally-mixed aerosol, there would thus be a particle size above which the average fluorescent signal would be greater than the threshold and below which the average fluorescent signal would be less than the threshold. This would make that size appear to be the true mode of the ensemble, but it would actually just be a reflection of the intrinsic fluorescent strength of the material within these particles. The 'true' diameter, so to speak, would be smaller than what it appears to be. Looking at a size-resolved average values of the FL signals for each FL channel would verify whether or not this data is in the regime. If not (and the signal strengths are sufficiently large relative to the applied threshold), this would add confidence to the reported mode diameter in Figure 8a. This is a general analysis issue for the fluorescent particle measurement community, and given that a paragraph of page 11 is devoted to comparing mode diameters between this and previous studies, I recommend this analysis.

Without knowing the identity of the particles and their resultant morphologies and whether their fluorophores are likely to be found on the surface or in the bulk of the particle, it is difficult to answer how size may influence fluorescent intensity. There is currently a lack of stable solid pure compound fluorescent calibrants to assess how particle size influences fluorescence (i.e., is there a surface area or volume dependence? Is there a maximum penetration depth?), but it not unreasonable to expect that fluorescence increases with particle size. This is an ongoing area of research, and is beyond the scope of this paper.

p9l17: "Cl2 appears to be. . .somewhat less fluorescent." What exactly do you mean by saying 'less fluorescent?' Table 3 indicates the mode diameter of the cluster is 1.9um compared to 2.5um for Cl1. Would a 1.9um Cl1 particle have the same fluorescent intensity as a 2.5 um Cl2 particle?

From table 3 of the paper, it can be seen that clusters 1 and 2 display similar characteristics, i.e., they mainly fluoresce in Fl1 with weak fluorescence in Fl2 and Fl3, however, the mean Fl1 intensity is greater for Cluster 1.  This is in contrast to cluster 3, which is mainly fluorescent in Fl3 and likely of different origin.  The similarities and strong correlation (p9|18) between clusters 1 and 2 suggests that they are of similar origin, with the difference in fluorescence being due to size, morphology or particle age.  We will clarify this in the revised manuscript.

As stated above, it is difficult to answer how size may influence fluorescent intensity.

p9l18: "Both clusters show similar fluorescent signatures to the clusters attributed to fungal spores by Crawford." How are the fluorescent signatures similar? In absolute intensity values? If that is the case, are the two instruments using the same detector gain settings, such that it would make sense to compare the intensities on an absolute scale? Or, are they similar in the relative strengths of channels Fl1-Fl2-Fl3? Even for relative differences between the channels, differences in gain settings would still be relevant in trying to compare this instrument's response with another. Further explanation and/or analysis on the spectral information collected by this WIBS should be provided to support the conclusion that these clusters represent fungal spores. There are other WIBS studies that have identified WIBS signatures for fungal spores as well (see Healy 2012 in Atm Env; Perring, 2015 in JGR; Hernandez, 2016 in AMTD) that would be worth comparing your results to, perhaps here or in section 3.5.2.

The signatures are both referenced to the FT + 3 standard deviation threshold representing an intensity of 0 as discussed earlier.  The cluster average values for this experiment and the BEACHON experiment (Crawford et al. 2014,2015), when compared, show that fluorescent signatures relative to the fluorescent detection threshold for BUNIAACIC cluster 1 and BEACHON cluster $Z_1$ are similar, i.e., both display strong fluorescence in Fl1 and moderately weak fluorescent in Fl2 and Fl3.  Both of these clusters also display a strong diurnal cycle with a dependency on relative humidity (see figure 2, which we will include in the revised manuscript as further evidence of this dependency on RH). This behaviour is consistent with that of emission of fungal spores (Hirst, 1953; Pringle et al., 2005; Elbert et al., 2007; Jones and Harrison, 2004).

Both datasets were collected with the same WIBS-3 using identical detector gain settings.  The WIBS-3 does not have a high and low gain mode as found in the WIBS-4 and WIBS-4A.

Direct comparison to other studies is not possible due to differences in detector gain (which currently cannot be calibrated) and the choice of excitation and detection wavebands.  Even comparing results between the same model of instrument with identical detector/filter configurations has been difficult, as shown in Hernandez et al., (2016).

[Figure]

Figure 2. Total particle number in clusters 1 and 2, plotted against relative humidity.

p9l19: I find the following statement confusing: "These clusters (referring to Cl1 and Cl2) contribute approximately 70% to the total FBAP concentration, with no significant diurnal variation." Yet there is a very strong diurnal signal in FBAP, and Cl1+Cl2 makes up 70% of FBAP. Is there a typo here, or am I misunderstanding the phrase 'with no significant diurnal variation' in Cl1+Cl2?

This is not a typo, just badly worded, and we apologise for the confusion. We meant to say that there was no variation in the 70% figure (i.e. there is a strong diurnal variation in Cl1+Cl2, but the make up 70% of FBAP regardless of time of day), but accept that the text is rather obscure. We will clarify this in the revised manuscript.

p9l25: A general comment on this section: the comparison of the HTDMA data made during this study with other previous work done in the same region (or similar regions) seems well done. However, there has been plenty of work done previously on submicron aerosol composition in this region, and there is very little discussion of your ACSM data within the context of this previous work. Please add some content (perhaps a paragraph) in this section comparing your ACSM results to other measurements that have been made here in the Amazon.

The following text has been added to P.9 L.25

"During BUNIAACIC, submicron non-refractory aerosol concentration shows significantly higher concentration (~2.5 µg m$^{-3}$) than observed at the remote sites in Central Amazonia in previous years during the wet season, ranging from 0.4 µg m$^{-3}$ (Artaxo et al., 2013) and 0.6 µg m$^{-3}$ (Andreae et al., 2015; Chen et al., 2009). Conversely, the concentration is significantly lower than reported during

the dry season (8.9 μg m$^{-3}$) (Andreae et al., 2015), as consequence of this transitional period not having extensive biomass burning activities, however with already reduced wet deposition due to reduced precipitation. Interestingly, despite the marked changes in ambient concentration, very little differences are observed in terms of relative contributions considering this and previous studies, being strongly dominated by organics (~80%), followed by sulphate and minor contribution of nitrate and ammonium (Andreae et al., 2015; Artaxo et al., 2013; Chen et al., 2009)."

p11l19: There are a number of studies not mentioned in this comparison section that the current manuscript would benefit from citing and discussing: -1. Poschl 2010: They attribute 80% of coarse-mode particles as primary biological particles. While those measurements were done with SEM, they seem to align with these results and should be mentioned. -2. Please also include in this paragraph how your results compare to PBAP modeling work that covers this region (e.g. Spracklen and Heald 2014). -3. A recent study on fungal spore measurements in the coarse mode, "Significant influence of fungi on coarse carbonaceous and potassium aerosols in a tropical rainforest." By Zhang and co-workers. They estimate fungal spore concentrations in a similar environment. There may be more studies. As this section is meant to compare your results to what has come before, a more thorough review of the literature should be done, and should not just be limited to aerosol fluorescence measurements as there are other ways of determining concentrations of airborne fungal spores.

We will include further discussion on these and other studies in the context of our work in this section.

p12l10: Similar to an earlier comment, it is not clear to me if there was no data recorded of biomass-burning influenced air, or if there was the influence of biomass burning but those data were flagged and removed. You write here ". . .the results here may reflect the transition between the two seasons, with periods consistent with each at different times (but without any influence from biomass burning)." The confusion arises because I am left wondering if the air sampled during this period is similar when you discount biomass burning influence, or if the air sampled here is similar partially because there is no biomass burning influence.

Pollution episodes, including biomass burning influences, were removed from the data prior to analysis, so that we could present the measurements of natural (biogenic) aerosol. We will clarify this in the conclusions section to avoid this confusion.

Technical corrections:

p6l27: This does not need to be a new paragraph.

We will modify the text accordingly.

Figure 1: Change "Particle number size distribution for the experiment" to "Particle number size distribution averaged over the entire measurement campaign" or something similar. Also, there are kappa and GF data here as well, which should also be mentioned in the caption.

We will change the caption in the revised manuscript.

Figure 2: Change caption to "The time-series of total particle counts (top panel) and particle number size distribution (bottom panel)." The order of what you list should go top to bottom, and with the multiple panel figures explicitly naming what is where reduces any possible confusion.

We will change the cation accordingly.

Figure 4: Can you make use of the entire range of the ROYGBIV colorscale? Almost all of the data is blue-ish/green, making use of the rest of the scale would make the data more visible here. Also given that this figure comes after a previous figure with many gaps, I would move the gaps statement ("Gaps are largely due. . .") up to Figure 2 or include this statement in each caption.

We will modify the colour-scale. In response to an earlier comment, we plan to signify gaps due to pollution events in these time-series figures with shading (or similar), and explain this in each caption.

Figure 5: I assume this is the case, but is the pie-chart for the average of all the data shown here? State this briefly in the figure caption.

This will be added to the figure caption.

Figure 9: "Mean growth factor for the dominant less hygroscopic mode" should be "Mean growth factor for the dominant, less-hygroscopic mode." Also typo with "agains."

We will correct the grammar and spelling in this cation.

Table 3: What are the units here? E.g. there should be units next to Cl1, Cl2, etc. What are the units of Asymmetry factor? (else a definition of what "Af" actually is should be provided somewhere in the text)

We will include the appropriate units in the table header. The asymmetry factor has arbitrary units, and we will define it in section 2.4.

References:

Andreae et al., 2015: The Amazon Tall Tower Observatory (ATTO): overview of pilot measurements on ecosystem ecology, meteorology, trace gases, and aerosols. Atmos. Chem. Phys. 15, 10723–10776, doi:10.5194/acp-15-10723-2015

Artaxo et al., 2013: Atmospheric aerosols in Amazonia and land use change: from natural biogenic to biomass burning conditions. Faraday Discuss. 165, 203–235, doi:10.1039/C3FD00052D

Brito et al., 2014: Ground-based aerosol characterization during the South American Biomass Burning Analysis (SAMBBA) field experiment. Atmos. Chem. Phys. 14, 12069–12083, doi:10.5194/acp-14-12069-2014

Chen et al., 2009: Mass spectral characterization of submicron biogenic organic particles in the Amazon Basin. Geophys. Res. Lett. 36, L20806. doi:10.1029/2009GL039880

Crawford et al., 2014: Characterisation 5 of bioaerosol emissions from a Colorado pine forest: results from the BEACHON-RoMBAS experiment, Atmospheric Chemistry and Physics, 14, 8559–8578, doi:10.5194/acp-14-8559-2014

Crawford et al., 2015: Evaluation of hierarchical agglomerative cluster analysis methods for discrimination of primary biological aerosol, Atmospheric Measurement Techniques, 8, 4979–4991, doi:10.5194/amt-8-4979-2015

Crawford et al., 2016: Observations of fluorescent aerosol–cloud interactions in the free troposphere at the High-Altitude Research Station Jungfraujoch, Atmos. Chem. Phys., 16, 2273-2284, doi:10.5194/acp-16-2273-2016

Elbert et al., 2007: Contribution of fungi to primary biogenic aerosols in the atmosphere: wet and dry discharged spores, carbohydrates, and inorganic ions, Atmospheric Chemistry and Physics, 7, 4569–4588, doi:10.5194/acp-7-4569-2007

Hernandez et al., 2016: Composite Catalogues of Optical and Fluorescent Signatures Distinguish Bioaerosol Classes, Atmos. Meas. Tech. Discuss., doi:10.5194/amt-2015-372 (in review)

Hirst, 1953: Changes in atmospheric spore content: diurnal periodicity and the effects of weather, T. Brit. Mycol. Soc., 36, 375–393, doi:10.1016/S0007-1536(53)80034-3

Jones and Harrison, 2004: The effects of meteorological factors on atmospheric bioaerosol concentrations–a review., The Science of the total environment, 326, 151–80, doi:10.1016/j.scitotenv.2003.11.021

Martin et al., 2010: An overview of the Amazonian Aerosol Characterization Experiment 2008 (AMAZE-08). Atmos. Chem. Phys. 10, 11415–11438, doi:10.5194/acp-10-11415-2010

Perring et al., 2015: Airborne observations of regional variation in fluorescent aerosol across the United States, J. Geophys. Res. Atmos., 120, 1153–1170, doi:10.1002/2014JD022495

Pöhlker et al., 2012: Autofluorescence of atmospheric bioaerosols – fluorescent biomolecules and potential interferences, Atmos. Meas. Tech., 5, 37–71, doi:10.5194/amt-5- 37-2012

Pöschl, U et al., 2010: Rainforest aerosols as biogenic nuclei of clouds and precipitation in the Amazon., Science (New York, N.Y.), 329, 1513–6, doi:10.1126/science.1191056

Pringle et al., 2005: The captured launch of a ballistospore, Mycologia, 97, 866–871, doi:10.3852/mycologia.97.4.866

Robinson et al., 2013: Cluster analysis of WIBS single-particle bioaerosol data, Atmos. Meas. Tech., 6, 337-347, doi:10.5194/amt-6-337-2013

Schneider et al., 2011: Mass-spectrometric identification of primary biological particle markers and application to pristine submicron aerosol measurements in Amazonia. Atmos. Chem. Phys. 11, 11415–11429, doi:10.5194/acp-11-11415-2011

Spracklen and Heald, 2014: The contribution of fungal spores and bacteria to regional and global aerosol number and ice nucleation immersion freezing rates, Atmos. Chem. Phys., 14, 9051-9059, doi: 10.5194/acp-14-9051-2014

Toprak and Schnaiter, 2013: Fluorescent biological aerosol particles measured with the Waveband Integrated Bioaerosol Sensor WIBS-4: laboratory tests combined with a one year field study, Atmos. Chem. Phys., 13, 225–243, doi:10.5194/acp-13-225-2013

Zhang et al., 2015: Significant influence of fungi on coarse carbonaceous and potassium aerosols in a tropical rainforest, Environ. Res. Lett., 10(3), doi:10.1088/1748-9326/10/3/034015

---

## Author Comment (AC2) · 10 May 2016

**Response to Anonymous Referee #2**

The paper 'Biogenic cloud nuclei in the Amazon' presented by Whitehead et al. contains a detailed compilation of different measurements during a 3-weeks intensive in the transition period between wet and dry season at a remote research station in the Amazon. The authors focused on different measurements of micro-physical, chemical and hygroscopic properties of the sub-micron aerosol particle population as well as the fluorescence of super-micron particles - a thoroughly interesting, comprehensive and significant data set. The collected data and shown results are relevant to the scientific community and contribute to a deeper understanding of the significance of (biogenic) aerosol particles for cloud properties and the formation of (mixed-phase) precipitation and hence the hydrological cycle in the Amazon.

The subject matter is clearly in the area of ACP. Nevertheless, I think several aspects concerning the data analysis and further technical issues need to be revisited carefully before the manuscript can be accepted for publication in ACP. Please find my major comments below.

We wish to thank the referee for taking the time for this thorough review of our manuscript. We address each of the comments below, detailing the changes made to the manuscript in response.

**General Comments:**

The manuscript shows an interesting but brief compilation of individual data sets, which are finally compared to previous studies. Since the whole data set comprises (as stated by the authors) a large variability e.g., for the total particle number concentration (100 - 800 cm-3, cf. Fig. 2), shape of the particle number size distribution (cf. Fig. 1), organic mass contribution measured by the ACSM (0.5 - 4 μg m-3, cf. Fig. 5), one would expect to find similar variability in GF or kappa. Nevertheless, GF and kappa are mainly discussed in terms of campaign averages and the applied color scale in Fig. 4 makes it hard to identify variability. Interestingly, the time series of GF does show clear episodes of stable conditions (cf. July 22th) versus episodes with higher variability (cf. July 23rd). Furthermore, during a short event on July 15th GF shows extraordinary high values (> 1.6), which is not discussed in the manuscript.

I suggest to carefully revisit the results section towards a more systematic and comprehensive analysis and discussion combining information from different measurements (particle number size distribution, total particle number concentration, hygroscopicity and chemical information).

In order to bring together the various measurements, we will include a greater discussion of the variability of GF and kappa (and modify the colour-scale in figure 4, also in response to a comment by referee #1), and a derivation of kappa from ACSM data to compare to those kappa from HTDMA and CCNc measurements.

The authors apply a hierarchical cluster analysis to the WIBS data, which is certainly a powerful technique to identify PBAB meta-classes. However, there is significant information missing about the input to the analysis and the corresponding discussion. This paragraph is not clearly outlined making it hard to follow the argumentation.

We refer the reviewer to our responses to comments made on this subject by referee #1. We will add a short description of the method to the revised manuscript. Complete information on this analysis technique and its implementation is available from Crawford et al (2015), to which we refer in the manuscript, and it is not practical to repeat it in full in this paper.

Finally, the title is very unspecific and does not clearly reflect the content of the paper.

We will change the title to "Biogenic cloud nuclei in the Central Amazon during the transition from wet to dry season".

I summarize more specific comments below.

**Specific comments:**

Section 2.1:

• first paragraph: The authors compare rainfall, temperature and humidity during their measurement period with AMAZE-08. Please specify the statement 'cooler and more humid'.

We will include numbers comparing the temperature and humidity between the two campaigns.

• second paragraph: This paragraph deals with detailed information on the location of the measurement site. Please consider to add a map. This would also be helpful for the discussion concerning the removal of pollution episodes.

We will include a map in the revised manuscript.

Section 2.5:

• The authors describe how they flag and remove pollution episodes from the entire data set. Last sentence: 'Approximately 28% of the HTDMA and CCNc data were removed in this way, with 5% of the data being flagged as possibly impacted by biomass burning and most of the rest due to the Manaus urban plume.'

• Why are only HTDMA and CCNc data removed? Additionally, data gaps in the shown figures have to be specified.

We will specify in the text how much ACSM data were removed due to flags. In addition we will include some shading (or similar) in each of the time-series figures signifying gaps due to pollution flags and add an explanation in each caption.

• I further suggest to consider to show a figure containing all geographical information including the mentioned Manaus bounding box.

We will include this in the map mentioned above.

Section 3.2:

• In section 2.5 the authors already introduce a 'cleaning procedure' to exclude pollution episodes. Does f60 show any correlation with the detected pollution events?

Previous studies in the Amazon have observed that a large fraction of the biomass-burning related organic aerosols do not present a significant f60 signal, due to long-range transport (Brito et al., 2014). As such, applying a f60 threshold would remove only fresh fires and not biomass burning emissions. We accept that the text isn't at all clear on this, and we will revise it to clarify this point. The statement in the current text simply says that the relatively low f60 confirms for us that there was no sign of local BB influence during measurements. We also refer the referee to our response to referee #1 on the same matter.

• p. 7, ll. 21: 'The mean f60 at TT34 in July 2013 was 0.19% ± 0.07%. This is well below 0.3%, which is considered to be the upper limit for background air masses not affected by biomass burning' Have the ACSM data been filtered? Is the mean value calculated after removing pollution events?

Yes, and we will add a note in the text clarifying this.

Section 3.4:

• p. 8, l. 18: 'mean total particle number concentration of FBAP ..' Do you mean the mean FBAP or the mean total particle number concentration?

We mean the mean total particle number concentration, and will correct the text in the revised manuscript.

• p. 8, l. 31: 'The observed night-time peak in FBAP number concentrations in fig. 7 is consistent with nocturnal sporulation driven by increasing RH' Where did you measure T and RH? Are the measurements collocated (below or above canopy)or part of the regular measurements at the research tower (if so, at which height)?

The RH was from the routine measurements from the top of the tower, and so was not collocated with the WIBS. We will add a note to the revised text.

• p. 9, l. 8: '. . . FBAP clearly dominates the particle number concentrations for Dp > 1 μm, however non-FBAP concentrations are higher for submicron particles': How robust is the characterization of the WIBS instrument? I wonder if this statement might be influenced by a decrease in sensitivity of the fluorescence signal. According to Crawford et al. (2015), the WIBS-4 has a 50% detection diameter at 0.8 μm. Please specify the 50% detection diameter of your instrument.

The instrument $D_{50}$ is 0.8 μm, we will revise §2.4 it include this information. The fluorescence response/collection efficiency is unknown for all UV-LIF instruments as there is a lack of an appropriate calibration/reference standard to perform such characterizations, as discussed in our response to referee #1. We will also modify the statement quoted here by the referee, to clarify that we mean the larger sub-micron particles (i.e. > 0.8 μm).

• p. 9, ll. 13: The authors apply a cluster analysis to the WIBS data without providing details on the data preparation and the precise input. According to the cited paper by Crawford et al. (2015), several steps are involved to filter the data before clustering. Did the authors apply exactly the same criteria? Even if so it is worth mentioning those criteria and the corresponding rejection rate in this manuscript.

The exact same method/criteria were applied in this analysis. We will revise §3.4 to clarify this. Approximately 15% of the single particle data was rejected based on this criteria, i.e., inclusion required D>0.8 μm, fluorescent in at least one channel and no detector saturation.

• p. 9, ll. 15: It is hard to follow the argumentation concerning the cluster analysis: 'Cl1 has previously been attributed to fungal spores (Crawford et al., 2014) based on comparison with other sampling techniques and the diurnal emission pattern (see fig. 7) with higher concentrations observed overnight' Was Cl1 attributed to fungal spores based on the observed diurnal cycle (in this publication) or on the mean values (of FL1-3, AF, size) of the corresponding cluster in Crawford et al., 2014?

The attribution of Cl1 to fungal spores was primarily based on the observed diurnal cycle and response to RH (see response to referee #1) and we will include a discussion of the RH dependence in the revised manuscript to clarify this. The similarity of the cluster centroids and the behaviour of the cluster to the work in Crawford et al., 2014 were presented as additional supporting information. We agree to clarify this in the revised manuscript.

• p. 9, ll. 20: 'The statistical parameters of each cluster are shown in table 3 for comparison. Together, these clusters contribute approximately 70% to the total fluorescent particle concentration, with no significant diurnal variation in this figure, suggesting that FBAP were dominated by fungal spores during this study.' Why does the hierarchical cluster analysis cluster only 70% of the data? Why is there no significant diurnal variation? And why does it in this case lead to the stated conclusion?

We agree that this section is not clear and will be revised. "Together, these clusters contribute approximately 70% to the total fluorescent particle concentration" refers to the sum of clusters 1 and 2, not the sum of all clusters, i.e., the clusters representative of fungal spores account for 70% of

the fluorescent population by concentration. The HCA method used here clusters all of the input data.

We meant to say that there was no variation in the 70% figure (i.e. there is a strong diurnal variation in Cl1+Cl2, but the make up 70% of FBAP regardless of time of day), but accept that the text is rather obscure. We will clarify this in the revised manuscript.

Section 3.5.1:

• p. 10, l. 28: 'The HTDMA derived _ from the Borneo experiment shows more hygroscopic aerosol than in Amazonia, as discussed above, however the CCNc derived values are more in line with those in Amazonia. This discrepancy has been noted previously and possible reasons for it discussed by Irwin et al. (2011) and Whitehead et al. (2014).' It would be interesting to discuss the findings of the mentioned papers in the context of the here observed discrepancy.

We will add a brief summary of the discussion from those paper in the revised manuscript.

Section 3.5.2:

• p. 11, l. 7: 'The median number concentration of FPAB observed below the canopy in this study was 372 l−1'. Unprecise – which study do you mean, Gabey et al. (2010) or this study?

This study. We will clarify this in the text.

• Concerning the observed discrepancies with Huffmann et al. (2012), the authors discuss instrumental issues, mixing effects related to strong vertical gradients and pbl development. I suggest to add a discussion about possible effects of wet deposition, since the measurements of Huffmann et al. (2012) were performed during the wet season.

We will include a couple of sentences in the revised manuscript discussing the possible role of wet deposition in the differences between these measurements.

• p. 11, l. 28: 'Diurnal variations between this study and that of Huffman et al. (2012) were similar, however Gabey et al. (2010) reported an additional increase in the afternoon in Borneo'. Unprecise – which measurement parameter increases?

In this paragraph we are discussing FBAP number concentrations. We will clarify this in the text.

**Technical issues:**

Please reference all your physical variables in the text and/or figure captions.

We will modify the captions / text appropriately.

Please do not use abbreviations like 'don't' (e.g., p. 11, l. 32).

We will modify the text accordingly.

Figure captions miss significant information:

Fig. 1:

• information on the derived GF and kappa is missing

We will add this information to the caption

• HTDMA, CCNc data comprise different measurement periods. Please specify that in the figure. Are these data averaged over the same time period?

HTDMA and CCNc data comprise the same measurement periods. We will clarify this.

Fig. 2:

• NCN - is this measured by the CPC or integrated from the size-resolved measurements?

NCN is integrated from the size-resolved measurements, and we will clarify this in the caption.

• please specify the data gaps

We will specify the data gaps according to pollution flags and/or instrument down-time as discussed in response to a previous comment.

Fig. 4:

• please specify the data gaps#

As above

• all other figures use GF(D/D0) instead of 'Growth Factor D/D0'

We will modify the label to GF(D/D0).

Fig. 5:

• please specify the data gaps

As above

• The unit is probably µg/m3

That is the unit specified in the axis label.

• What is the collection efficiency for the ACSM data?

The following text has been added to P.7L.24:

"The instrument collection efficiency was calculated to be 1 during BUNIAACIC, through the comparison of the mass concentration of species measured by the ACSM and MAAP (black carbon) with the integrated mass of the SMPS. Further details of the method are given by Brito et al. (2014) and Stern et al. (in preparation).

Fig. 6:

• Ntot refers to the size range of the WIBS, make sure that there is no confusion with the term 'total counts' in Fig. 2

We will specify this in the figure caption

Fig. 7:

• Ntot refers to the size range of the WIBS, make sure that there is no confusion with the term 'total counts' in Fig. 2

We will specify this in the figure caption

• please add information about the sensor height and position for T and RH

We will add this information to the figure caption.

• °C

This is correct

Fig. 8 a & b:

• you use Dp instead of Dp

This will be corrected for the revised manuscript

• unit of dN/dlog dp is wrong

This will be corrected for the revised manuscript

Fig. 9: 'Irwin et al., (2011)'

References:

• page 16, line 15: lower case initials: 'Wiedensohler, Arana'

We will correct this

• page 17, line 3: full name instead of initials: 'Anna Stefaniak'

We will correct this

References:

Brito et al., 2014: Ground-based aerosol characterization during the South American Biomass Burning Analysis (SAMBBA) field experiment. Atmos. Chem. Phys. 14, 12069–12083, doi:10.5194/acp-14-12069-2014

Crawford et al., 2014: Characterisation 5 of bioaerosol emissions from a Colorado pine forest: results from the BEACHON-RoMBAS experiment, Atmospheric Chemistry and Physics, 14, 8559–8578, doi:10.5194/acp-14-8559-2014

Crawford et al., 2015: Evaluation of hierarchical agglomerative cluster analysis methods for discrimination of primary biological aerosol, Atmospheric Measurement Techniques, 8, 4979–4991, doi:10.5194/amt-8-4979-2015

Gabey et al., 2010: Measurements and comparison of primary biological aerosol above and below a tropical forest canopy using a dual channel fluorescence spectrometer, Atmospheric Chemistry and Physics, 10, 4453–4466, doi:10.5194/acp-10-4453-2010

Huffman et al., 2012: Size distributions and temporal variations of biological aerosol particles in the Amazon rainforest characterized by microscopy and realtime UV-APS fluorescence techniques during AMAZE-08, Atmospheric Chemistry and Physics, 12, 11997–12019, doi:10.5194/acp-12-11997-2012

Irwin et al., 2011: Size-resolved aerosol water uptake and cloud condensation nuclei measurements as measured above a Southeast Asian rainforest during OP3, Atmospheric Chemistry and Physics, 11, 11 157–11 174, doi:10.5194/acp-11-11157-2011

Whitehead et al., 2014: A meta-analysis of particle water uptake reconciliation studies, Atmospheric Chemistry and Physics, 14, 11 833–11 841, doi:10.5194/acp-14-11833-2014

---

## Author Comment (AC3) · 10 May 2016

**Response to Anonymous Referee #3**

This manuscript reports the results of aerosol measurements taken place in the Amazon basin during the wet-to-dry transition period. The measurements include particle size distributions, hygroscopicity, and fluorescent biological aerosol particle concentrations, and are compared to the previous measurements. The results are important and interesting, especially since there are few previous studies in that environment. However, it is not clear to me why the authors choose to remove pollution episodes from this dataset and how this "clean" dataset provides a "unique contrast (page 2, line 10) to the wet-season data?" In fact, the observed particle total number concentrations and hygroscopicity as well as chemical composition are quite similar to those observed during the wet season. The WIBS-3 results are different but also largely because the measurements were done within the canopy. To me, the removed data are really the key feature of the transition period, meaning influences but not as strong as the dry season. It is important to add that analysis as a contrast. The authors should also pay attention to the manuscript preparation guidelines for authors provided by the journal (http://www.atmospheric-chemistry-andphysics.net/for_authors/manuscript_preparation.html). I recommend this manuscript be published after the following comments are addressed.

We wish to thank the referee for taking the time to thoroughly review our manuscript. The main focus of this paper is on the natural (biogenic) aerosol at this location during the transition from wet to dry seasons. The contrast with the wet season is due to the difference in meteorology. We will replace "contrast" with "comparison" in the quoted text to avoid confusion. We agree that it would be useful to consider biomass burning influenced air masses as well; however as we state in section 2.5, most of the removed data was due to pollution from Manaus (which is not unique to any time of year). The data flagged as *possibly* influenced by biomass burning accounts for only 5% of the data, which we did not consider sufficient to allow for a good comparison.

We address each of the referee's other comments below.

Specific comments:

(1) A 5-paragraph abstract seems unnecessary for this paper. Some of the details may be removed and the key points need to be summarized more concisely.

We will shorten the abstract as far as possible.

(2) Page 3, line 19-20: Please provide the relative humidity and temperature for both campaigns.

We will include this information in the revised manuscript.

(3) Page 4, line 30; Page 5, line 32: Do you mean "polystyrene latex spheres (PSL)" for both cases? What sizes have you used for the calibration? Do the uncertainties for growth factor derived from HTDMA vary by D0? What do you mean "blue fluorescent latex spheres"? Please clarify. Also, since different kinds of diameters are described in the paper, the authors should specify the diameter type in the text and figures.

We do mean polystyrene latex spheres (PSL), and will clarify this, as well as including the sizes used for each calibration. Sub-micron particles are measured as mobility diameter, while the WIBS measured the optical diameter; we will clarify this. We will clarify how we used the blue fluorescent latex spheres, and add that they were manufactured by Polysciences Inc., PA, USA, and Duke Scientific Corp., CA, USA.

(4) Page 6, line 4-5: Do you mean "some of the PBAP are detected by WIBS"? Please clarify and give examples.

We mean some PBAP won't necessarily be detected by the WIBS, as discussed by Gabey et al (2010) and Huffmann et al (2012). We will clarify this and expand the discussion in the revised manuscript.

(5) Section 2.5: It is not clear to me which flag was applied to which dataset and whether if the flag was properly set. The authors should provide clear information about the data processing and have consistency among datasets.

First of all, Figures 2, 4, and 5 look like having different gaps (lack of clear description in the graphs and figure captions about the gaps).

The flags were applied in the same way to the HTDMA, CCNc, ACSM and size data. We will make this clearer in section 2.5 of the revised manuscript. In addition, we will specify the periods in each of these figures where data were removed due to pollution flags (by shaded areas, or similar), and explain this more clearly in the captions. Some additional gaps were due to instrument down-time. Again, we will clarify this in the relevant captions.

Second, the back trajectories at all altitudes from 0 to 4000 m.a.s.l were used for the identification of pollution episodes (page 6, line 17). However, most sampling was taken at 39 m (10 m above canopy) and WIBS was operated on the ground level.

Issues can arise with back-trajectories initiated at ground level, due to the effects of the terrain on air flow, and the greater chance of the trajectory intersecting the ground. To overcome this, we investigated the trajectories at several heights. In terms of the pollution flags, the results were largely the same at the 0 – 2 km levels and very little influence from the upper level flow at 4 km. We will clarify this in the revised manuscript.

Third, it was said that data sampled for local wind direction of 270°-340° were flagged as potential generator contamination (line 27). But in line 31-32, the authors said that 5% of the removed data were potential biomass burning and the rest were Manaus plume. Then, which part is due to generator contamination?

In fact, there were no instances of flagged generator contamination during the measurement periods in this study. We will explain this in the revised manuscript.

Finally, in line 29-30, significant increases in black carbon concentration and particle number concentration were used as the second criteria of data removal. The question is "are there periods with such significant increases but not flagged by the back trajectories passed over Manaus, fire zone, or by wind direction for generator plumes?" If so, when and why? If not, the former (increases) is enough for identifying the pollution episodes.

There were no other increases in black carbon or particle number concentration outside the flagged periods, but we couldn't have known this when we started our analysis of pollution events. In addition, the exercise in flagging by back trajectories and wind sectors allows us to identify the nature of the pollution event (Manaus plume or biomass burning influence). We believe that this level of redundancy in flagging data for pollution events is important to ensure that we have done this as rigorously as possible, without inadvertently removing otherwise good data.

(6) Section 3.1: Both paragraphs said that the observed particle number size distributions are similar to the ones measured in the dry season (i.e., effected by biomass burning). However, the data are supposed to represent background conditions because of the removal of pollution episodes.

And they do: it is the shape of the distribution that is similar, while the number concentrations in this study are somewhat lower than during the dry season. We will add a sentence (with values) explaining this.

(7) Page 7, line 21-23: The author should clarify that the ACSM data (Fig. 5) do not cover the entire measurement period (Figs 2 and 4). "in July 2013" is inaccurate. Have the excluded periods flagged by biomass burning shown elevated f60?

Previous studies in the Amazon have observed that a large fraction of the biomass-burning related organic aerosols do not present a significant f60 signal, due to long-range transport (Brito et al., 2014). As such, applying a f60 threshold would remove only fresh fires and not biomass burning emissions. We accept that the text isn't at all clear on this, and we will revise it to clarify this point. The statement in the current text simply says that the relatively low f60 confirms for us that there was no sign of local BB influence during measurements. We also refer the referee to our response to referee #1 on the same matter.

(8) Page 7, line 28: What are the definitions of hydrophobic, less or more hygroscopic mode (page 10, line 2) in terms of growth factor? Are their definition consistent in literatures (e.g., for the comparisons done in page 10, line 1-12?

The terms in quotation marks are as defined in the cited literature, however we will define the growth factor ranges to make comparison easier.

(9) Page 7, line 29-30: What does the "local anthropogenic influence" stand for? What is "this distribution (i.e., . . .)"?

Here we are speculating that the hydrophobic mode is due to some unknown local anthropogenic source, and will insert the word "unknown" to make it clear. Then we refer to the growth factor distribution, and will insert the term "growth factor".

(10) Page 8, line 1-5: Increased growth factor with particle dry diameter can be explained by many possibilities (it doesn't have to be greater sulfate contribution at larger diameter; organic material at different diameter may different as well). Without careful analysis, I think it is hard to demonstrate that the observations here reflect similar size-resolved chemical information to the previous studies. And the particle number size distributions observed in this study are indeed different from what was observed in previous wet-season studies as described in Sect. 3.1.

We agree with the referee that careful analysis is needed before we can draw this conclusion. We will modify this paragraph to say that higher sulphate concentration is a possible explanation.

(11) Page 9, line 15-24: The analysis here is confusing and needs clarifications. It was said first that C11 is attributed to fungal spores and C12 remain unclassified. Then why "both clusters show similar fluorescent signatures to the clusters attributed to fungal spores"? Aren't all the three classes distinct in fluorescent signatures (line 15)?

We accept that this isn't clear and we agree to revise this section to the following:

"Cl1 like particles have previously been attributed to fungal spores (Crawford et al., 2014) based on comparison with other sampling techniques and the diurnal emission pattern (see fig. 7) with higher concentrations observed overnight. Cl2 appears to be a distinct sub-class of Cl1 which is less fluorescent in FL1. Cl2 shows similar behaviour to, and correlates strongly (r2 = 0.86) with Cl1, hence both have been combined in fig. 7. Both clusters show similar fluorescent signatures to the clusters attributed to fungal spores by Crawford et al. (2014, 2015)."

Second, in line 21, it was said that "these clusters . . ., with no significant diurnal variation in this figure, suggesting that FBAP were dominated by fungal spore during this study." Does "these" mean C11+C12 or C11+C12+C13? Don't C11 and C12 show nighttime increase in Fig. 7? Finally, if C13's concentration is low, what about the residuals in the cluster analysis (meaning Fig. 7 showed a difference of hundreds in number concentration between FBAP and C11+C12)? What does the "insufficient data" mean in line 24?

We are referring to Cl1+Cl2, and we will clarify this in the revised manuscript. Cl1 and Cl2 do show a nighttime increase; we meant to say that there was no variation in the 70% figure, but accept that the text is rather obscure. We will clarify this in the revised manuscript.

The residual difference between $N_{FBAP}$ and $N_{Cl1}+N_{Cl2}$ is a result of rejecting saturated particles from the cluster analysis input.

Insufficient data refers to a lack of additional supporting data which could be used to infer the origin of Cl3, e.g., response to rainfall may infer that the particles are bacterial. We will clarify this in the revised manuscript.

(12) Page 10, line 9 and line 12: What does "strong diurnal cycles" mean? Daytime peak? Please clarify.

The "strong diurnal cycles" refers to an increase in the fraction of moderately hygroscopic particles, and we will clarify this in the revised manuscript.

(13) Page 10, line 31-32: What about the removed data? Do those data show very different results compared to the "clean" conditions? Also it is important to explain why the particle concentrations and hygroscopic properties are similar to those during the wet season but the particle size distributions are similar to those observed in the dry season (my comment #6, Sect. 3.1).

The removed data was mostly flagged as pollution from Manaus, which is of no interest to this study, while the data flagged as possibly influenced by biomass burning made up less than 5% of the total, which we considered insufficient to provide a significant result. We will expand the discussion on the differences / similarities with the wet season.

(14) Page 11, line 11-12: What kind of meteorological conditions? Need a reference or example to support this hypothesis. Also, what are "other locations"? Please specify.

We refer to the wetter, more humid conditions of the wet season favouring sporulation, and we will include references in support. We will specify in the revised manuscript the locations for each citation here.

Technical remarks:

Page 3, line 18-19: Revise "the AMAZE-08 campaign saw 370 mm fall" and move the reference to the end.

We will revise as necessary

Page 3, line 27: Revise "local time was UTC – 4 hours".

We will revise as necessary

Page 3, line 31: "RH" has not been defined yet.

We will define RH here

Page 4, line 21 and 25: Properly revise "dry sizes" since the DMA selects a band of the electric mobility not just one size.

We will use appropriate terminology in the revised manuscript

Page 4, line 28-29: "a bubble flowmeter" is an improper description. Also, shouldn't be "Gillibrator-2"?

We will change this to "air flow calibrator", and "Gillibrator-2".

Page 5, line 23-26: What is NADH? What do you mean "3 fluorescence channels"?

We will define NADH here. The 3 fluorescence channels are already defined in the preceding text.

Page 6, line 4: Add "as" after "termed" and revise the later part of the sentence.

We believe "termed" is used correctly here, but will revise the end of the sentence.

Page 7, line 3 and later text: "fig. " should be "Fig. ".

We will revise accordingly

Page 7, line 8: "particle counts" should be "particle number concentrations".

We will revise accordingly

Figure 5. Remove frame. Figure 5 appeared earlier than Fig. 4.

We will remove the frame, and change the order of the figures.

Page 7, line 27: Should be "in the range of 1.2 to 1.4" (the word "of" is missing).

We will revise accordingly

Page 8, line 8-9: Check the grammar for " at larger diameters _ _ . . . and _ _ 0.18 around the accumulation mode. " SI units should be used, and units in the denominator should be formatted with negative exponents.

We will revise the grammar as necessary. Kappa has no units.

References:

Brito et al., 2014: Ground-based aerosol characterization during the South American Biomass Burning Analysis (SAMBBA) field experiment. Atmos. Chem. Phys. 14, 12069–12083, doi:10.5194/acp-14-12069-2014

Crawford et al., 2014: Characterisation 5 of bioaerosol emissions from a Colorado pine forest: results from the BEACHON-RoMBAS experiment, Atmospheric Chemistry and Physics, 14, 8559–8578, doi:10.5194/acp-14-8559-2014

Crawford et al., 2015: Evaluation of hierarchical agglomerative cluster analysis methods for discrimination of primary biological aerosol, Atmospheric Measurement Techniques, 8, 4979–4991, doi:10.5194/amt-8-4979-2015

Gabey et al., 2010: Measurements and comparison of primary biological aerosol above and below a tropical forest canopy using a dual channel fluorescence spectrometer, Atmospheric Chemistry and Physics, 10, 4453–4466, doi:10.5194/acp-10-4453-2010

Huffman et al., 2012: Size distributions and temporal variations of biological aerosol particles in the Amazon rainforest characterized by microscopy and realtime UV-APS fluorescence techniques during AMAZE-08, Atmospheric Chemistry and Physics, 12, 11997–12019, doi:10.5194/acp-12-11997-2012

---

## Referee Report (RR1)

Referee #1 response to revised paper - Whitehead et al "Biogenic cloud nuclei in the Central Amazon during the transition from wet to dry season"

The revised manuscript has been improved beyond the original, and ultimately I recommend the article for publication.

There are a few issues from my initial review that I feel warrant addressing or were not fully addressed in the authors' response, which I have detailed below:

1.  Asymmetry factor

While the authors provided a reference to Crawford et al, addressing the need for some attempt at defining this quantity, the inclusion of Af does nothing for the manuscript. It is not discussed, as far as I could tell, in any way. Possibly ways of discussing this would include: what do the Af values for each cluster mean on an absolute scale? How do the values of the clusters compare to eachother? How do they compare to other studies? What does a given value of Af…mean? As it stands, without any discussion of this quantity at all, all mentions of asymmetry factor should be removed.  Of course, if it can be discussed and shown to be a quantity worth discussing in the manuscript, then by all means keep it in.

2. Crawford et al

In updated discussion within the manuscript on fluorescent signatures, the authors briefly describe some of the challenges in comparing fluorescent particle measurements across instruments. In the response to one of my comments, it was mentioned that the WIBS-3 used in this study is the same instrument, operated in the same way, that was used in Crawford et al. Given how many times the Crawford study is cited in this manuscript, and within the context of the discussion about the difficulty of comparing measurements across studies, please explicitly state that indeed this instrument is the same model and operated the same as in Crawford et al.

3. Figure 1

I appreciate the inclusion of an updated Figure 1 in the response to my review. Why not include it in the final manuscript though? As I stated earlier, this plot quickly and effectively communicates the difference in scale between the sub and super-micron modes in this location. I think it is important for readers to see a presentation like this given how much of the manuscript is dedicated to discussion of the super-micron. Contrary to what was stated in the response to my review, it does add to the paper.

---

## Author Response (AR2)

We once again thank the referees for reviewing our manuscript for a second time and providing further suggestions. Our brief responses are below, firstly to referee #1:

1. Asymmetry factor

While the authors provided a reference to Crawford et al, addressing the need for some attempt at defining this quantity, the inclusion of Af does nothing for the manuscript. It is not discussed, as far as I could tell, in any way. Possibly ways of discussing this would include: what do the Af values for each cluster mean on an absolute scale? How do the values of the clusters compare to eachother? How do they compare to other studies? What does a given value of Af…mean? As it stands, without any discussion of this quantity at all, all mentions of asymmetry factor should be removed. Of course, if it can be discussed and shown to be a quantity worth discussing in the manuscript, then by all means keep it in.

We have added a brief discussion of how the Af further suggests the clusters Cl1 and Cl2 may be fungal.

2. Crawford et al

In updated discussion within the manuscript on fluorescent signatures, the authors briefly describe some of the challenges in comparing fluorescent particle measurements across instruments. In the response to one of my comments, it was mentioned that the WIBS-3 used in this study is the same instrument, operated in the same way, that was used in Crawford et al. Given how many times the Crawford study is cited in this manuscript, and within the context of the discussion about the difficulty of comparing measurements across studies, please explicitly state that indeed this instrument is the same model and operated the same as in Crawford et al.

We have added this statement to the revised manuscript.

3. Figure 1

I appreciate the inclusion of an updated Figure 1 in the response to my review. Why not include it in the final manuscript though? As I stated earlier, this plot quickly and effectively communicates the difference in scale between the sub and super-micron modes in this location. I think it is important for readers to see a presentation like this given how much of the manuscript is dedicated to discussion of the super-micron. Contrary to what was stated in the response to my review, it does add to the paper.

We have now included this figure, and referred to it from the section on size distributions.

And to referee #3:

A minor suggestion is to link the meteorological difference to the results when comparing to the wet-season data. For example, as the paper stated, the conditions were hotter and less humid. Are we expecting more biogenic SOA formation under such conditions compared to the wet season? Are there any possible explanations for the greater loadings of submicron organic matter and the dominated accumulation mode etc. observed in this study?

We have added a paragraph to the end of section 3.5.1 to discuss this.

All the technical corrections were also applied.

**Biogenic cloud nuclei in the Central Amazon during the transition from wet to dry season**

J. James D. Whitehead[1], E. Eoghan Darbyshire[1], J. Joel Brito[2], H. Henrique M. J. Barbosa[2], I. Ian Crawford[1], R. Rafael Stern[3], M. Martin W. Gallagher[1], P. Paul H. Kaye[4], J. James D. Allan[1], H. Hugh Coe[1], P. Paulo Artaxo[2], and G. 
[revised manuscript text omitted]